# A Bayesian framework for emergent constraints: case studies of climate sensitivity with PMIP

Martin Renoult[1], James Douglas Annan[2], Julia Catherine Hargreaves[2], Navjit Sagoo[1], Clare Flynn[1], Marie-Luise Kapsch[3], Qiang Li[4], Gerrit Lohmann[5], Uwe Mikolajewicz[3], Rumi Ohgaito[6], Xiaoxu Shi[5], Qiong Zhang[4], and Thorsten Mauritsen[1]

[1]Department of Meteorology, Bolin Centre for Climate Research, Stockholm University, Stockholm, Sweden
[2]Blue Skies Research Ltd, Settle, United Kingdom
[3]Max-Planck Institute for Meteorology, Hamburg, Germany
[4]Department of Physical Geography, Bolin Centre for Climate Research, Stockholm University, Stockholm, Sweden
[5]Alfred Wegener Institute, Helmholtz Centre for Polar and Marine Research, Bremerhaven, Germany
[6]Japan Agency for Marine-Earth Science and Technology, Yokohama, Japan

**Correspondence:** Martin Renoult (martin.renoult@misu.su.se)

**Abstract.**

In this paper we introduce a Bayesian framework, which is explicit about prior assumptions, for using model ensembles and observations together to constrain future climate change. The emergent constraint approach has seen broad application in recent years, including studies constraining the equilibrium climate sensitivity (ECS) using the Last Glacial Maximum (LGM) and the mid-Pliocene Warm Period (mPWP). Most of these studies were based on ordinary least squares (OLS) fits between a variable of the climate state, such as tropical temperature, and climate sensitivity. Using our Bayesian method, and considering the LGM and mPWP separately, we obtain values of ECS of 2.7 K (0.6–5.2, 5–95 percentiles) using the PMIP2, PMIP3 and PMIP4 data sets for the LGM, and 2.3 K (0.5–4.4) with the PlioMIP1 and PlioMIP2 data sets for the mPWP. Restricting the ensembles to include only the most recent version of each model, we obtain 2.7 K (0.7–5.2) using the LGM and 2.3 K (0.4–4.5) using the mPWP. An advantage of the Bayesian framework is that it is possible to combine the two periods assuming they are independent, whereby we obtain a tighter constraint of 2.5 K (0.8–4.0) using the restricted ensemble. We have explored the sensitivity to our assumptions in the method, including considering structural uncertainty, and in the choice of models, and this leads to 95% probability of climate sensitivity mostly below 5 K and only exceeding 6 K in a single and most uncertain case assuming a large structural uncertainty. The approach is compared with other approaches based on OLS, a Kalman filter method and an alternative Bayesian method. An interesting implication of this work is that OLS-based emergent constraints on ECS generate tighter uncertainty estimates, in particular at the lower end, an artifact due to a flatter regression line in case of lack of correlation. Although some fundamental challenges related to the use of emergent constraints remain, this paper provides a step towards a better foundation of their potential use in future probabilistic estimation of climate sensitivity.

# 1 Introduction

In recent years, researchers have identified a number of relationships between observational properties and a future climate change, which was not immediately obvious a priori, but which exists across the ensemble of global climate models (GCMs) (Allen and Ingram, 2002; Hall and Qu, 2006; Boé et al., 2009; Cox et al., 2018) participating in the Climate Model Intercomparison Project (CMIP). These relationships are generally referred to as 'emergent constraints' as they emerge from the ensemble behaviour as a whole rather than from explicit physical analysis.

Such emergent constraints have been broadly used to constrain properties of the Earth's climate system which are not easily or directly observable. These are usually presented in probabilistic terms, mostly based on Ordinary Least Squares (OLS) methods. For example, studies have explored the constraint on equilibrium climate sensitivity (ECS), which is the global mean equilibrium temperature after a sustained doubling of $CO_2$ over pre-industrial levels, using model outputs from the Paleoclimate Model Intercomparison Project (PMIP) (Hargreaves et al., 2012; Schmidt et al., 2014; Hopcroft and Valdes, 2015; Hargreaves

and Annan, 2016). Because of their relatively strong temperature signal, paleoclimate states like the Last Glacial Maximum (LGM) and the mid-Pliocene Warm Period (mPWP) are often considered as promising constraints for the ECS (Hargreaves et al., 2012; Hargreaves and Annan, 2016), in particular at the high end.

Almost all emergent constraint studies have used OLS-based methods to establish the link between variables in the model ensembles. However, whether ECS or another climate parameter was investigated, the theoretical foundations for the calculations

have not previously been clearly presented. An additional problem arising from this is the resulting difficulty in synthesising estimates of climate system properties generated by different statistical methods with different, and often not explicitly introduced, assumptions. These methods include OLS but also alternative Bayesian approaches such as estimates of the climate sensitivity using energy-balance models (Annan et al., 2011; Aldrin et al., 2012; Bodman and Jones, 2016).

Two recent papers have also addressed the question of emergent constraints in different ways. Bowman et al. (2018) pre-

40 sented a hierarchical statistical framework which went a long way to closing the gap in theoretical understanding of emergent constraints. Conceptually, it is very similar to a single step Kalman filter, where the iteration process is avoided to only keep a single updating of a prior into a posterior. Specifically, it uses the model distribution approximated as a Gaussian as a prior, which is then updated using the observation to a posterior. However, such prior and the underlying assumptions attached to it could be seen as a restrictive choice to impose on the climate sensitivity area of research. In particular, most of the posterior

values would lie in the range covered by the ensemble of models if the observed value is either uncertain and/or close to the prior mean. This is a direct consequence of the joint probability distribution produced by the Kalman filter, which in the case of joint Gaussian distributions, will produce a tighter posterior Gaussian distribution. Because of that, it does not appear to correspond to the choice which is usually made, albeit implicitly.

Another Bayesian statistical interpretation of emergent constraints has been recently presented by Williamson and Sansom

(2019) who extended the standard approach to account for more general sources of uncertainty including model inadequacy. A key aspect of their approach is that they set a prior on the observational constraint rather than the climate system parameter(s)

that we are primarily interested in this study, i.e. the climate sensitivity. Thus, their prior predictive distribution for the climate system parameter is not immediately clear and may not be so easily specified as in the approach we explore here.

We present an alternative Bayesian linear regression approach in which the regression relationship is used as a likelihood model for the problem. This allows for the prior over the predictand to be defined separately from and entirely independently of the model ensemble and emergent constraint analysis. Thus the likelihood arising from the emergent constraint could be used to update a prior estimate of the predictand that arose from a different source.

In Section 2 we provide an overview of the concept of emergent constraints, the previous methods used for these analyses, introduce the Bayesian framework as well as the models and data employed in the paper. Section 3 describes the results, starting with analysis of models and data from the Paleoclimate Intercomparison Project (PMIP) Phases 2 and 3 for the LGM and mPWP, that have previously been analysed for an emergent constraint on climate sensitivity (Schmidt et al., 2014; Hargreaves and Annan, 2016). We then incorporate some CMIP6/PMIP4 model outputs that have been made available to us for these periods to illustrate how these outputs fit into the same analysis. We also use the LGM and mPWP outputs to demonstrate how the method allows independent emergent constraints to be combined. Finally, we discuss the influences of the prior and of model inadequacy on climate sensitivity in Sections 3.5 and 3.6, respectively.

## 2 Methods

The general method of "emergent constraints" seeks a physically plausible relationship in the climate system between two model variables in an ensemble of results from different climate models. Consequently, an observation of one measurable variable (such as past tropical temperatures) could be used to better constrain the other investigated variable, usually unobserved and difficult to measure (such as climate sensitivity). This idea has been used in climate science to estimate quantities of interest such as snow albedo feedback (Hall and Qu, 2006), future sea ice extent (Boé et al., 2009; Notz, 2015), low-level cloud feedback (Brient et al., 2016), and the equilibrium climate sensitivity (Hargreaves et al., 2012; Schmidt et al., 2014; Cox et al., 2018). Although the unobserved variable is usually taken as a future variable, the emergent constraints theory can be used with two variables within the same time frame, as long as the relationship is plausible. A summary of several different emergent constraints on climate sensitivity was made by Caldwell et al. (2018). This approach using emergent constraints is meaningful only if we believe that reality satisfies the same relationship, and it was not observed purely by chance in the model ensemble. There is a risk in searching for such relationships in a small ensemble that we may find examples which are coincidental, with no real predictive value (Caldwell et al., 2014). Spurious relationships could also be found because of model limitations (Fasullo and Trenberth, 2012; Grise et al., 2015; Notz, 2015).

In this study, we focus on the relationship between equilibrium climate sensitivity, defined here as $S$, and the temperature change in the tropics which is observed at the Last Glacial Maximum (LGM) and the mid-Pliocene Warm Period (mPWP), defined as $T_{\text{tropical}}$. We posit that a relationship between climate sensitivity and temperature change is physically plausible, as we expect the long-term quasi-equilibrium temperature to be mainly influenced by radiative forcing, and in many model

ensembles, variations in climate sensitivity have been dominated by tropical feedbacks, mostly arising from low-level clouds (Bony et al., 2006; Vial et al., 2013).

## 2.1 Ordinary Least Squares

The most widely-used approach to emergent constraint analysis is to find an observable phenomenon that exhibits some relationship to the parameter of interest, and use this as a predictor in a linear regression framework. The ordinary least squares (OLS) method has been widely used because of its simplicity, and so we also use it here as a starting point for comparison with alternative statistical methods. In the context of constraining climate sensitivity, the parameter of interest (i.e. the ECS) is considered as a predicted variable (Hargreaves et al., 2012; Schmidt et al., 2014; Hargreaves and Annan, 2016). This may be written as

$$S = \gamma \times T_{\text{tropical}} + \delta + \zeta \tag{1}$$

where $S$ is the climate sensitivity, $\gamma$ and $\delta$ two unknown parameters, $T_{\text{tropical}}$ the temperature anomaly averaged over the tropical region for the given paleo-time interval, and $\zeta$ the residual term which is drawn from a Gaussian distribution $N(0, \sigma^2)$ and which accounts for deviations from the linear fit. When we use this approach, the unknown constants of the linear fit are estimated via ordinary least squares (OLS) using the $(T^i_{\text{tropical}}, S^i)$ pairs representing the model ensemble (here $i$ indexes the models) and then the equation is used to predict the true value of $S$ for the climate system, based on the observed value $T^o_{\text{tropical}}$. A confidence interval for the predictor variable can be generated by accounting for uncertainties in the fit and in the observed value through a simulation of an ensemble of prediction as was demonstrated by Hargreaves et al. (2012). This procedure makes the assumption that reality satisfies the same regression relationship as the models, i.e. is likely to be at a similar distance from the line as the model points are.

Integrating the intrinsically frequentist confidence intervals obtained from regression methods used for OLS estimates into a Bayesian framework is challenging. One issue is the misinterpretation of frequentist confidence intervals as Bayesian posterior credible intervals. The former is the representation of the number of random intervals to contain the true interval bounds (at 90% confidence, this would lead to 90 out of 100 random intervals to contain the true bounds), while Bayesian credible interval is an interval which we believe (with the given probability) to contain the truth. For instance, if there is an observed $T_{\text{tropical}} = 1$ K, with an assumed Gaussian observational uncertainty of $\sigma = 0.25$ at one standard deviation, then stating that there is a close-to-95% probability of having the true value of the parameter within the interval 0.5–1.5 K is a Bayesian credible interval interpretation. However, the latter is a common interpretation of frequentist-based studies. This confusion has inherent drawbacks on the analysis of posterior outputs, as shown in various fields of science (Hoekstra et al., 2014) and more recently, for climate sensitivity computations (Annan and Hargreaves, 2019). Williamson and Sansom (2019) have presented a Bayesian interpretation of this approach using reference priors on $\psi$, as defined by Cox et al. (2018) as a metric of global mean temperature variability, and the regression coefficients. However, this approach does not appear to readily allow for the use of any arbitrary prior distribution for $S$ which may either be desired for comparison with other research, or else have arisen

through a previous unrelated analysis. The Bayesian linear regression approach that we introduce in the next section avoids these problems.

## 2.2 Bayesian framework

The (subjective) Bayesian paradigm is based on the premise that we use probability distributions to describe our uncertain beliefs concerning unknown parameters. We use Bayes' Theorem to update a prior probability distribution function (pdf) for the equilibrium climate sensitivity via

$$P(S|T^o_{\text{tropical}}) = \frac{P(T^o_{\text{tropical}}|S)P(S)}{P(T^o_{\text{tropical}})} \tag{2}$$

where $P(S|T^o_{\text{tropical}})$ is the posterior estimate of $S$ after conditioning on the geological proxy data $T^o_{\text{tropical}}$, $P(S)$ is the prior and $P(T^o_{\text{tropical}})$ is a normalisation constant. The likelihood $P(T^o_{\text{tropical}}|S)$ is a function that takes any value of $S$ and generates a probabilistic prediction of what we would expect to observe as $T^o_{\text{tropical}}$ if that value was correct. The use of the Bayesian paradigm requires us to create such a function. Using the principles of emergent constraint analyses in which a linear relationship between these two parameters, which was seen in the GCM ensemble, is believed to apply also to reality, it is natural to use the regression relationship

$$T_{\text{tropical}} = \alpha \times S + \beta + \epsilon \tag{3}$$

where here, $\alpha$, $\beta$ and $\sigma$, the standard deviation of $\epsilon$ as $\epsilon \sim N(0, \sigma^2)$, are three a priori unknown parameters. Note that this reverses the roles of predictor and predictand compared to the OLS-based approach (Eq. 1). It implies that $S$ is able to give a prediction of $T_{\text{tropical}}$, with a given uncertainty. This is physically plausible, as $S$ is considered as one of the best metrics to represent temperature change. In particular, $S$ is often diagnosed in climate models from abrupt and sustained quadrupling of $CO_2$ from pre-industrial conditions (4x$CO_2$), which usually leads to weak non-linearity similar to what shall be observed from LGM or mPWP climate dynamics. Therefore, it is possible to use 4x$CO_2$-computed $S$ of climate models to predict $T_{\text{tropical}}$, assuming $\epsilon$ as a representation of all processes not related to $S$.

Choosing $S$ as the predictor (Eq. 3) will cause some differences to the inference of posterior $S$ compared to the OLS-based approach introduced in Eq. 1. The plausibility of the existence of an emergent constraint between $S$ and $T_{\text{tropical}}$ is independent of the method chosen. Whether $T_{\text{tropical}}$ is a predicted or predictor variable, or whether the applied method uses OLS or Bayesian statistics, the methods estimate different unknown parameters to investigate a similar assumed relationship within the model ensemble; and so it is expected that these different methods will yield similar but not identical results. This was previously argued in the context of hierarchical statistical model for emergent constraints by Tingley et al. (2012). The Bayesian approach with $S$ as the predictor is appropriate for emergent constraint analyses thanks to its transparency and handling of uncertainties. This has been explored by Sherwood et al. (2020), and is also investigated in this study. Thus, here we explore the implementation of the Bayesian method for emergent constraint analyses, to models and data that have already been investigated with alternative methods (Hargreaves et al., 2012; Hargreaves and Annan, 2016).

The three parameters $\alpha$, $\beta$ and $\sigma$ in Eq. 3 are conditioned on the model ensemble defined by its pairs of $(T^i_{\text{tropical}}, S^i)$ (with $i$ indexing the models). We estimate them via a Bayesian linear regression (BLR) procedure, which requires priors to be defined over these parameters. Consequently, the likelihood $P(T_{\text{tropical}}|S)$ for a given $S$ (as required by Eq. 2) is an integration over the posterior distribution of $T_{\text{tropical}}$ predicted by the regression relation (convolved with observational uncertainty where appropriate) and conditioned on the model ensemble through $\alpha$, $\beta$ and $\sigma$.

In this way, we create a statistical model that can generate a predictive pdf for the tropical temperature change at the LGM or at the mPWP $P(T_{\text{tropical}}|S)$, for any given sensitivity. There is a structural difference between this approach and that of Eq. 1, in that here the residual uncertainties $\epsilon \sim N(0, \sigma^2)$ represent our inability to perfectly predict the tropical temperature anomaly arising from a given sensitivity, and are probabilistically independent of the latter rather than the former variable. The issue here is not a matter of which regression line is 'correct', but rather how, given the model ensemble, we can create a plausible likelihood model for $P(T_{\text{tropical}}|S)$.

It is important to note that Eq. 3 and the conditioning of parameters on the model ensemble only relates to the generation of the likelihood. The emergent constraint calculation itself is then a second step that uses this likelihood to calculate the posterior of interest $P(S|T^o_{\text{tropical}})$ (Eq. 2). To apply the emergent constraints theory, it is required to insert a geological observation $T^o_{\text{tropical}}$ estimated through proxy data, and obtain the likelihood $P(T^o_{\text{tropical}}|S)$ which leads to the posterior $P(S|T^o_{\text{tropical}})$ by Bayesian updating. We perform this step through a simple importance sampling algorithm by approximating $P(T_{\text{tropical}} = T^o_{\text{tropical}}|S)$. That is, for any given sensitivity $S$, we can calculate the probability of the observation of tropical temperature that we have, as the composition of the predictive pdf for actual tropical temperature, together with the uncertainty associated with the observation itself. The emergent constraint theory is thus applied with a 2-stage Bayesian process, including in first the BLR and in second, a Bayesian updating.

A prior belief on climate sensitivity ($P(S)$) in the Bayesian updating process, and on the parameters of the regression model in the BLR process, has to be assumed. There is no clearly uncontested choice for prior distribution for climate sensitivity. However, Annan and Hargreaves (2011) argued that a Cauchy distribution has a reasonable behaviour with a long tail to high values, but unlike the uniform prior, does not assign high probability to these values. Thus we adopt this prior for our main analyses. In section 3.5 we test the sensitivity of the results to this choice and compare the results obtained using Gamma and uniform prior distributions. Priors for the parameters of the regression model are chosen with reference to the specific experiment and are intended to represent our reasonable expectation that models do indeed generate a regression relationship as described.

An additional issue, that was briefly mentioned above, is that we may like to consider the probability that reality is qualitatively and quantitatively distinguishable from all models. This issue, which was explicitly argued in the context of emergent constraint analysis by Williamson and Sansom (2019), seems reasonable since all models do share a theoretical heritage and certain limitations. However, this issue remains challenging to quantify. It has not been considered in most previous studies which also makes it difficult to compare. We investigate this issue in Section 3.6. Whilst the proposed resolution remains preliminary and although the concept is promising for understanding emergent constraints, we decide to omit it for the bulk of our analysis to enable more direct comparisons with previous studies.

The Bayesian method is more explicit than the standard OLS approach, as the prior assumptions have to be given by the user. This transparency leads to more freedom and control of the statistical model. Moreover, it has a reduced sensitivity to outliers as the prior on the regression coefficients provides a form of regularisation. This should lead to lower variance in the results compared to results with wider priors on the parameters, particularly with small model ensembles.

Additionally, the Bayesian method allows the user to add multiple lines of evidence by updating the chosen prior for $S$. The method for combining independent constraints is reasonably simple, as it only requires us to calculate and store the posterior of the first emergent constraint analysed, and use this distribution as the prior for the second emergent constraint. Thus it is a direct form of sequential Bayesian updating. This process results in a posterior distribution which will generally be narrower than either of the two posteriors that would have been generated from either of the emergent constraints separately. Although it may be tempting to simply combine all emergent constraints in this way, it is necessary to also consider possible dependencies between the uncertainties in the different emergent constraints before this can be done with confidence (Annan and Hargreaves, 2017).

It is not clear if observational errors have always been adequately accounted for in previous emergent constraints research. Our approach provides a natural framework for this, as the likelihood can include the uncertainty of the observational process as we have done. Recent studies have investigated Bayesian ways of integrating uncertainties on proxy reconstructions into global temperature field (e.g. Tierney et al. (2019)). For the sake of comparison with Hargreaves et al. (2012); Schmidt et al. (2014); Hargreaves and Annan (2016), we use the reconstructions and observational errors adopted in these studies which are based on multiple linear regressions and model-proxy cross-validation. However, we have ignored uncertainties in the calculation of the model values of $S$ and $T_{\mathrm{tropical}}$ as, while they are poorly quantified, we believe them to be too small to materially affect our result. In fact, it has been argued for the case of the mPWP that observational errors on $S$ and $T_{\mathrm{tropical}}$ are small compared to the structural differences responsible of the dispersion of the points around the regression line and thus can be neglected (Hargreaves and Annan, 2016).

## 2.3 Kalman Filter

Bowman et al. (2018) recently presented a new interpretation of emergent constraint analysis. Their framework is essentially a two-dimensional ensemble Kalman Filtering approach in which the prior, represented by the model ensemble, is updated according to the observation, using the Kalman equations which approximates all distributions by a multivariate Gaussian (Kalman, 1960). The Kalman equations are given by

$$K = P^{\mathrm{f}} H^{\mathrm{T}} \left( H P^{\mathrm{f}} H^{\mathrm{T}} + R \right)^{-1} \tag{4}$$

$$x^{\mathrm{a}} = x^{\mathrm{f}} + K \left( z - H x^{\mathrm{f}} \right) \tag{5}$$

$$P^{\mathrm{a}} = (I - KH) P^{\mathrm{f}} \tag{6}$$

where $x$ is the mean, $P$ its covariance, $z$ the observations with associated observational uncertainty covariance matrix, $R$ and $H$ the operator that maps the model state onto observations. The superscripts f and a by convention refer to the forecast (i.e. the prior in this work) and analysis (the posterior) respectively. While in many applications, such as numerical weather

prediction, this method is applied in an iterative fashion with the analysis being used as the starting point of the next forecast, here it is only applied once as a way of implementing Bayesian updating to our prior in order to generate the posterior.

Here we only have two dimensions for the Gaussian, these being the scalar predictor (e.g. sensitivity) and predictand (e.g. tropical temperature change). While this approach is a natural and attractive option in many respects, it has the specific draw-back (in the context of this work) of using the distribution of model samples as a prior (for both mean and covariance). Existing literature on emergent constraints does not make this assumption and this could be seen as a limiting aspect of the method, as it implies that the model ensemble is already a credible predictor even before consideration of the observational constraint. Some implications of this approach are that the posterior estimate will be equal to the model distribution in the case that no observational constraint exists, either because there is in fact no relationship between observation and predictand, or else when the observational uncertainty is excessively large. The use of a Gaussian prior based on the ensemble range also means that it is difficult for the method to generate posterior estimates that include values significantly outside the model range, even in the case where the observed value is outside the model spread. We present results generated with a Kalman filter in Section 3.1 for comparison with our main analysis.

## 2.4 Climate Models and Data

The Bayesian method may be applied to any emergent constraint. In this study, we use the model outputs and data syntheses arisen from phases 2 and 3 of PMIP (Braconnot et al., 2007; Haywood et al., 2011; Harrison et al., 2014), as well as the few available models of the phase 4 (Haywood et al., 2016; Kageyama et al., 2017), summarised in Table 1. The Last Glacial Maximum (19–23 ka) corresponds to the period of the last ice age where ice sheets and sea ice had their maximum extent. Due to its temporal proximity, relative abundance of proxy data, and substantial radiative forcing anomaly, the LGM is widely considered one of the best paleoclimate intervals for testing global climate models and has been featured in all of the PMIP consortium experiments. A representation of several model LGM simulations compared to the SAT reconstruction of Annan and Hargreaves (2013) is shown in Fig. 1–(a).

Previous results from PMIP2 showed a significant correlation between LGM tropical temperatures and climate sensitivity in the models (Hargreaves et al., 2012), although the equivalent calculation for the PMIP3 models found no significant correlation (Schmidt et al., 2014; Hopcroft and Valdes, 2015). These two similar sized ensembles with contrasting characteristics are a good test-bed for exploring the properties of the different methods. For the tropical temperature anomaly relative to pre-industrial we use a value from Annan and Hargreaves (2013), for 20° S to 30° N, a $T_{\text{tropical}}^o$ of -2.2 K with a Gaussian observational uncertainty of $\pm$ 0.7 K (5–95% confidence interval). Several data compilations are presently in development as part of PMIP4, but these have yet to be integrated into a global temperature field so revising the temperature estimate from Annan and Hargreaves (2013) is a topic for future work.

Interest in the mPWP (2.97–3.29 million years ago) as a more direct analogy for future climate change, has grown during the past years. This is the most recent period with a sustained high level of greenhouse gases and concomitant warmth relative to the pre-industrial period, however, the data are more sparse and uncertain. In Fig. 1–(b), the sea-surface temperature anomaly of different climate models which performed a mPWP simulation is displayed, as well as the PRISM3 SST reconstruction

(Dowsett et al., 2009). Previous results for this period from the Pliocene Model Intercomparison Project (PlioMIP) experiment, which was part of PMIP3, indicated a fairly strong correlation between tropical temperature and climate sensitivity in the models, but the confidence with which this can be used to constrain climate sensitivity was low due to high uncertainty in various observationally derived components as well as various compromises in the way the protocol was formulated (Hargreaves and Annan, 2016). For the mPWP, a tropical temperature anomaly of $0.8 \pm 1.6$ K (5–95% interval) is taken from Hargreaves and Annan (2016) for 30° S to 30° N, assuming the largest 5–95% uncertainty showed in that work. The reconstruction used here is the PRISM3 (Pliocene Research, Interpretation and Synoptic Mapping) SST anomaly field as described in Dowsett et al. (2009).

The Last Interglacial (127 ka, referred as lig127k in CMIP6) and the mid-Holocene (6 ka) are part of the PMIP simulations and also relatively warm climates. The forcings are, however, seasonal and regional in nature, mostly influencing the patterns of climate change. The global change in temperature and the global climate forcing are both very small, and this coupled with the large uncertainty in paleoclimate data makes these intervals poor candidates for constraining climate sensitivity. We do not explore these intervals further here.

Climate sensitivity has various definitions and there are also a number of different ways of approximating the value in climate models that have not been run to equilibrium. For PMIP3 LGM the model values are mostly based on the regression method of Gregory et al. (2004), but for the models which contributed to PMIP2 LGM and PlioMIP the exact definition and derivation used in each case is not always clear in the literature. In order to make comparisons with previous work, here we use the same values as those used in Hargreaves et al. (2012), Schmidt et al. (2014) and Hargreaves and Annan (2016) with two exceptions to ensure that only one value of sensitivity is used for identical versions of the same model across different experiments. Specifically, for FGOALS-g2 we use the value of 3.37 K (Yoshimori, pers. comm.) for both PMIP3 LGM and PMIP3 PlioMIP, and for HadCM3 we use 3.3 K (Randall et al., 2007) for both PMIP2 LGM and PMIP3 PlioMIP. Previous values used by Hargreaves and Annan (2016) for PMIP3 PlioMIP were 3.7 K for FGOALS-g2 (Zheng et al., 2013) and 3.1 K for HadCM3 (Haywood et al., 2013). These changes are minor compared to the ensemble range of climate sensitivity and thus, they have no significant effect on the posterior outputs.

In addition to the already published results from PMIP2 and PMIP3 we add to our ensembles the results that are currently available from PMIP4 in section 3.3. While the LGM protocol (Kageyama et al., 2017) remains very similar to that in previous iterations of PMIP, the mPWP protocol (Haywood et al., 2016) has more significant differences which address several of the limitations of the previous version. Most importantly, PlioMIP2 seeks to represent a specific quasi-equilibrium climate state in the past rather than representing an amalgamation of different warm peak climates as had been the case for PlioMIP1. A priori we are therefore less confident about combining the results from PlioMIP1 and PlioMIP2 and do so mostly to indicate where the new models lie in the ensemble and to highlight the potential for future research in this area once more model results based on the PlioMIP2 protocol become available.

## 3    Applications and Results

In order to apply the Bayesian Linear Regression and compute the likelihood $P(T_{\text{tropical}}|S)$, several priors have to be established as initial conditions. Specifically, for both the LGM and the mPWP we use Eq. 3 as the basis for our likelihood function. The prior expectations of the three unknown parameters $\alpha$, $\beta$ and the standard deviation of the residual $\epsilon$, referred to as $\sigma$, need to be defined. The relative complexity of the likelihood function with three a priori unknown parameters requires the use of a sampling method for computational efficiency. In this study, we use the Markov Chain Monte Carlo (MCMC) method NUTS as described by Hoffman and Gelman (2014). The NUTS method is also included in the MCMC python package PyMC3 (Salvatier et al., 2016) which is applied here. The approach is alternatively described as a conjugate priors problem using the R package spBayes (Finley et al., 2013, 2014), described in Appendix A, and leads to similar results.

Depending on the strength of the correlation among the dataset, one could expect a sensitivity of the regression to the choice of prior parameters. In the following sections, we first describe the physical arguments behind the choice of priors over $\alpha$, $\beta$ and $\sigma$, and then present the outputs of the BLR for both the PMIP2 and PMIP3 dataset of the LGM and the PlioMIP1 dataset of the mPWP. Then, we include the CMIP6 data in the Bayesian framework for both paleo intervals, and present an approach of combining the two emergent constraints. Finally, we explore the sensitivity of the Bayesian approach to the choice of priors over the climate parameter of choice (i.e. the climate sensitivity) and to the hypothetical inadequacy of climate models.

### 3.1    The Last Glacial Maximum

From consideration of energy balance arguments and fundamental physical properties, such as the response of Earth to an increase of $CO_2$, we have a prior expectation of a relationship between sensitivity and global LGM temperature anomaly (e.g. Lorius et al., 1990), and model experiments of Hargreaves et al. (2007) as well as simple physical arguments about the spatial distribution of forcing suggest that this relationship may be most clearly visible when we focus on the tropical region. While the total negative forcing at the LGM is roughly twice as large as the positive forcing that would be caused by a doubling of $CO_2$, the temperature response at low latitudes is generally expected to be lower than the global mean, due to polar amplification and the related presence of high latitude ice sheets. Thus we might reasonably expect the tropical temperature change at the LGM to be roughly equal to the global temperature rise under a doubling of $CO_2$. It would also be unexpected if the correlation had the opposite sign to that based on simple energy balance arguments, such that a more sensitive model had a lower temperature change at the LGM. However we cannot justify imposing a precise constraint on the slope and therefore our choice of prior for $\alpha$ is $N(-1, 1^2)$. As for $\beta$, we expect the regression line to pass close to the origin, as a model with no sensitivity to $CO_2$ would probably have little response to any other forcing changes, especially in the tropical region where the influence of ice sheets is remote. However, we do not expect a precise fit to the origin and therefore, the prior chosen for $\beta$ is $N(0, 1^2)$. Finally, we chose a wide prior for $\sigma$, a Half Cauchy with a scale parameter of 5. The Cauchy is fairly close to uniform for values smaller than the scale parameter, decaying gradually for higher values.

Deviations from the regression line may be due to different efficacies of other forcing components, especially ice sheets or dust. To take into account the uncertainty on the strength of the response, we performed two additional analyses where the

prior response was smaller ($\alpha$ defined as $N(-0.5, 1^2)$) and larger ($\alpha$ defined as $N(-2, 1^2)$). We do not see much difference in the results using the three priors over $\alpha$: the difference is approximately 0.2 K of climate sensitivity for both the upper and lower percentiles quoted, giving us confidence in our choice of $N(-1, 1^2)$. The computed 5–95% posterior climate sensitivity ranges for different values of $\alpha$ are summarised in Table 2.

The MCMC algorithm samples the posterior distribution of regression parameters which is represented by the ensemble of predictive regression lines in Fig. 2. This ensemble is used to infer the climate sensitivity following the Bayesian inference approach using the geological reconstruction of the LGM tropical temperature. The posterior distributions of $S$ are computed using a truncated-at-zero Cauchy prior with a peak of 2.5 and a scale of 3, which corresponds to a wide 5–95% prior interval of 0.5–28.7 K. Such a prior was used previously by Annan et al. (2011) because it has a long tail, allowing for a substantial probability of having high climate sensitivity while still maintaining some preference for more moderate values. However, the sensitivity of Bayesian statistics to the choice of prior has often been noted. Thus, two alternative priors, including the widely used uniform prior, and their corresponding posterior distributions, are investigated in Section 3.5.

To test the robustness of the method and also to compare it with the statistical methods used in previous studies, three cases are investigated in which we use different combinations of the available model ensembles. The results are shown in Fig. 2 and Table 2.

For the PMIP2 ensemble, the correlation between tropical temperature and climate sensitivity was found to be reasonably strong and in this study the resulting 5–95% range for inferred climate sensitivity is 1.0–4.5 K (Fig. 2–(b)). The range is slightly better constrained at the lower end than the 0.5–4 K from Hargreaves et al. (2012), however we have used the revised value for the LGM tropical anomaly of -2.2 ± 0.7 K rather than the value of -1.8 ± 0.7 K that was used by Hargreaves et al. (2012). The Bayesian-inferred value is similar to the OLS-inferred method with the revised version (Table 2), giving confidence on the proximity of both methods in case of high correlation.

When all the models of PMIP2 and PMIP3 (see Table 1) were considered jointly the correlation became weaker and the corresponding 5–95% range generated by the Bayesian method is 0.7–4.8 K (Fig. 2–(d)). Schmidt et al. (2014) obtained 1.6–4.5 K using a similar ensemble although in that case multiple results obtained from the same modelling centre were combined by averaging. Using the OLS method on our ensemble and generating predicted values, we obtain a 5–95% range of 1.4–4.6 K. The Bayesian method generates a wider range here, particularly at the lower end, as the correlation is weaker and the prior starts to influence the posterior.

Finally, we consider the PMIP3 models in isolation. For this ensemble no correlation is found so for the Bayesian method the result is heavily dependent on our prior assumptions. We obtain a 5–95% range here of 0.7–5.5 K (Fig. 2–(f)). Applying the OLS-based prediction method on the PMIP3 dataset gives a 5–95% range of 1.3–5.6 K. As previously argued for the combination of PMIP2 and PMIP3, the latter method produces a tighter posterior range at the lower end. In the absence of a correlation, the Bayesian method relaxes to the prior, whereas the predictions obtained via the OLS method are heavily influenced by the range of the ensemble. Additionally, as previously argued in Section 2.2, the differences in posterior 5–95% range between Bayesian and OLS-based approaches are partly connected to choosing $S$ as predictor or predicted, respectively. The impact of such choice will be even bigger as the correlation gets weaker, since the difference between the respective

error parameters $\epsilon$ and $\zeta$ will increase. However, we emphasise that this does not suggest that either range is closer to reality. Although the comparison between methods with predictor or predicted $S$ should get more complex from a philosophical point of view as $\epsilon$ differs from $\zeta$, we stipulate that both ranges can be considered as valuable information regarding $S$ within a climate and emergent constraint framework.

The Kalman filtering approach presented by Bowman et al. (2018) has not previously been used for emergent constraint analyses in paleoclimate research. Thus, we also use this method to explore both PMIP2 and the combination of PMIP2 and PMIP3 (Fig. 3). With the same geological reconstruction value, and a prior 5–95% range (based on the PMIP2 GCM ensemble) of 1.7–4.5 K, a posterior range of $S$ of 1.8–4.1 K is inferred. By combining the PMIP2 and PMIP3 models, the prior 5–95% range becomes 2.0–4.5 K and the posterior range is 2.2–4.2 K. The increase in lower bound in these calculations is the largest change compared to our Bayesian linear regression method. However, this is strongly forced by the underlying assumptions of a Kalman filter (Section 2.3) which uses the model ensemble as a prior, making it difficult to compute a posterior range outside of the model range, in particular when the observed value is considered as excessively uncertain. Thus, although the Kalman filtering method could be interesting, we do not consider it further, as we stipulate its assumptions are too restrictive for the question of emergent constraints and therefore can not be a relevant method in its current form to efficiently assess $S$ and, in particular, its uncertainty.

### 3.2 The mid-Pliocene Warm Period

As for the LGM, priors parameters have to be defined to perform the BLR with the mPWP data. In principle these may be different to those used for the LGM experiment, since the total positive forcing of the mPWP is not as large as the negative forcing of the LGM, but in practice we have adopted the same priors for our base case, apart from the obvious sign change for $\alpha$. We performed the same sensitivity experiments as for the LGM, with three different priors over $\alpha$: $N(1, 1^2)$, $N(0.5, 1^2)$, $N(2, 1^2)$. There was only a small difference between the results using the three priors: the differences at the 5th percentile being less than 0.1 K and the differences at the 95th percentile being approximately 0.3 K (see Table 2). Regarding $\beta$ and $\sigma$, there is no physical reason for their priors to be substantially different than the ones chosen for the LGM. Thus, a $N(0, 1^2)$ prior for $\beta$ is selected and the same prior for $\sigma$ as for the LGM analysis is chosen.

The Bayesian inference method applied above for the LGM model outputs is now applied on the mPWP model outputs (Fig. 4). With less abundant models and less well-constrained temperature data, we prefer to assume large uncertainties in the mPWP SST reconstruction ($0.8 \pm 1.6$ K, 5–95% confidence). We adopt the Cauchy prior on climate sensitivity as for the LGM analysis (5–95% interval of 0.5–28.7 K) and compute a 5–95% interval for the ECS of 0.5–5.0 K for the PlioMIP1 dataset. Similar to the results for the LGM, the predictions via OLS method (Hargreaves and Annan, 2016) resulted in a slightly narrower 5–95% range than the Bayesian method (1.3–4.2 K, assuming 1.6 K of uncertainty on the data).

### 3.3 Inclusion of CMIP6 / PMIP4 data

The ongoing PMIP4 experiments have produced LGM and mPWP (PlioMIP2) simulations. Here we add those results to our ensembles. There are four model runs available for the LGM and five for the mPWP (see Table 2) on 1 May 2020.

For the LGM we have previously combined the PMIP2 and PMIP3 results, and the protocol for PMIP4 is not very different. If we combine all three ensembles we obtain a 5–95% range for the ECS of 0.6–5.2 K using the Bayesian method (Fig. 5–(b)). The ensemble size is now 18, but we note that this includes several models coming from the same modelling centres. Past studies have investigated the proximity of models with hierarchical trees (Masson and Knutti, 2011; Knutti et al., 2013) and the influence of their dependency on statistical methods (Annan and Hargreaves, 2017). Thus, although we believe such dependencies exist in the ensemble, it is in reality difficult to quantify and correct for this. How to deal with this possible duplication of information is therefore a subjective decision. In Schmidt et al. (2014) it was taken into account by averaging the results from models from the same modelling centre. Here we take an alternative approach of including only the latest version of each model. This gives an ensemble size of 11 models (Table 2) and a 5–95% climate sensitivity range of 0.7–5.2 K with the Bayesian method. The range here is relatively wide and close to the range computed with the ensemble of PMIP2, PMIP3 and PMIP4. This is due to the removal of almost all PMIP2 models in this restricted ensemble, which leaves mainly the poorly correlated PMIP3 ensemble and the ensemble of PMIP4 together.

For PlioMIP1 and PlioMIP2 the situation is a little more complex as the protocol has been redesigned to represent a specific interglacial state rather than a generic warm climate, referred to as a "time slab" in the PlioMIP protocol. Thus there could be a different regression relationship for these two ensembles. However, when we plot the PlioMIP1 ensemble members (Fig. 5–(d)) we see that they do not look different to the PlioMIP2 ensemble members. The straight combination of PlioMIP1 and PlioMIP2 gives an ensemble range of 14 models and we computed a 5–95% range of 0.5–4.4 K. Including only the most recent versions of models results in an ensemble size of 11 models (Table 2) and generates a nearly identical 5–95% climate sensitivity range of 0.4–4.5 K with the mPWP simulation. Thus, for this period the inclusion of the PlioMIP2 models allows for a tighter constraint at the upper bound, much aided by the larger spread of $S$ in these new models.

### 3.4 Combining multiple constraints

As described in section 2.4, the mPWP and the LGM are very different climates. If the observational data are generated by unrelated analyses, we may be able to consider the two lines of evidence to be independent, and combine them using Bayes theorem to create a new posterior which is likely to be narrower than that arising from either analysis alone. Assuming that the uncertainties arising from the mPWP and the LGM analyses are independent of each other may be plausible as the proxy reconstructions use different observations and analyses to estimate both the tropical temperatures and the other variables that act as boundary conditions for the model experiments. Moreover, modelling uncertainties that influence the regression analysis are expected to arise from rather different sources, such as the response to ice sheets and a cold climate in one case, versus the influence of a warmer climate in the other. Having said that, model biases influencing the simulation of one climate change may also influence the other, which means that if similar models occur in both ensembles, this could lead to dependencies. Using Bayes theorem to combine the constraints means that it is not necessary for the same set of models to be used for each ensemble but, as we can see from Table 1, a few models do occur in both ensembles.

It is straightforward to first compute the posterior estimate of $S$ from the LGM analysis as previously described, and then use this as a prior for the mPWP analysis. Priors over the regression coefficients are considered independent between the two

analyses. Because of the issues discussed above, we perform an analysis using both ensembles of latest model versions in the LGM and the mPWP as described in section 3.3. The posterior of the LGM is used as the prior for the mPWP analysis and the resulting posterior from this process has a narrower 5–95% interval for $S$ of 0.8–4.0 K (Fig. 6).

A logical extension of the approach would be to apply it to the ensemble of models of CMIP, where multiple emergent constraints exist for the same models. In theory, this should be possible as long as the investigated relationships are physically plausible. This goes beyond the scope of our study, which uses the paleoclimates as an example for the method, and is left for future research.

## 3.5 Alternative Priors on sensitivity

A major strength of the Bayesian analysis developed here is the way that the prior on the parameter of interest, here climate sensitivity, can easily be specified independently of all other aspects of the analysis. A uniform prior for $S$ has been widely used (e.g. Tomassini et al., 2007; Aldrin et al., 2012). However, it has also been argued that such prior could give an unrealistically high weight to high climate sensitivity (Annan and Hargreaves, 2011). Here we test our method with the commonly-used uniform prior U[0;10] which has a 5–95% range of 0.5–9.5 K. The resulting posterior 5–95% range for climate sensitivity is 0.8–5.0 K when analysing the LGM PMIP2 models only, and 0.6–5.4 K with the LGM PMIP2 and PMIP3 models together. These posteriors are wider than the ranges previously computed with a Cauchy prior, particularly for the case of combining PMIP2 and PMIP3 where the correlation is rather weak, in which case the prior has a higher influence. These results are shown in Fig. 7. Due to the questions which have arisen over the use of a uniform prior and the fact that it has an infinite integral, unless bounded arbitrarily as done here, we also perform a comparison with an alternative prior which features a decaying tail and a finite integral. For this purpose, a Gamma prior is chosen with a shape parameter of 2 and a scale of 2, which corresponds to a similar 5–95% prior range of 0.7–9.5 K. The posterior computed 5–95% range is 1.0–4.5 K for LGM PMIP2 models and 0.9–4.8 K for the combination of PMIP2 and PMIP3, which is very close to the one computed with the Cauchy prior. Although the Bayesian paradigm will inevitably involve such subjective choices, the sensitivity of the results to a sensible choice of prior appears to be low as long as a reasonable correlation exists in the ensemble.

## 3.6 Model Inadequacy

As previously explored and described by Williamson and Sansom (2019), we investigate the probability that all models deviate in a systematic way from reality to a certain extent, mainly because of computational limitations and their shared technical heritage. Statistically, this issue is best described by the terminology that while the models are considered 'exchangeable' with each other, they are not exchangeable with reality. Williamson and Sansom (2019) provide further discussion on this point. In our methodology, this can simply be accounted for by considering that the regression prediction of $S$ for reality has a larger residual than that arising for the models themselves:

$$T_{\text{tropical}}^t = \alpha \times S^t + \beta + \epsilon^*, \tag{7}$$

where the superscript $t$ indicates here that we are referring to the truth (i.e. the real climate system) and $\epsilon^*$ has the distribution N(0,$\sigma^{*2}$) for some $\sigma^{*2} > \sigma^2$. There can be various reasons why such an inadequacy, represented as $\epsilon^*$ in Eq. 7, may be thought to exist. Models all share a common heritage and theoretical basis, which is certainly incomplete even if not substantially wrong, and computational constraints limit their performance. Particularly in the paleoclimate context, there may be biases in the experimental protocol and differences in number of feedbacks included in the different model systems, e.g. interactive vegetation and prognostic dust. Such errors would lead to reality being some distance from the model regression line, even if the models were otherwise perfect. Such issues are relevant to both the LGM, where there are significant uncertainties relating to dust and vegetation effects, and the mPWP where even the GHG forcing is somewhat uncertain, and furthermore where the older simulations are designed as a general representation of interglacial warm periods rather than a specific quasi-equilibrium climate state.

However, while we may anticipate reality deviating further from the regression line, it is difficult to quantify such deviation. Here, we perform two sensitivity tests where we define $\sigma^{*2} = (2\sigma)^2$, that is to say the distribution for the residual term $\epsilon^*$ is defined as N(0,$(2\sigma)^2$) for our predictions. We consider that this corresponds to a rather large inadequacy term. To compare with our previous analysis, we investigate the effect of the model inadequacy using the data set of PMIP2 and PMIP3 combined for the case of the LGM, and the data set of PlioMIP1 for the case of the mPWP. For the LGM, the 5–95% posterior range computed after doubling $\sigma$ is 0.5–5.8 K, while the 5–95% posterior range for the mPWP is 0.5–5.4 K. When we consider the 'latest model version' approach outlined in Section 3.3 and take the same approach of doubling the estimated residual, the 5–95% posterior ranges increase to 0.5–6.3 K for the LGM and a 5–95% posterior range of 0.4–5.0 K for the mPWP. Thus these sensitivity tests typically involve a change of around half a degree to the upper bound obtained, while having much less influence on the lower bounds in these examples.

## 4   Conclusions

Past climates are relevant sources of information on the properties of the climate system, specifically the equilibrium climate sensitivity, due to the quasi-equilibrium changes in response to external forcing, which are of similar magnitude to the projected future climate changes. In this study, we have described a new statistical method based on Bayesian inference to approach the question of emergent constraints. We believe this method provides a reasonable representation within the Bayesian paradigm of the underlying structure of emergent constraint principles. This Bayesian method is designed to be as explicit and flexible as possible. Previous work using ordinary least squares usually applied implicit assumptions. Because of these assumptions, predictions obtained via OLS tend to generate tight posterior ranges, particularly on the lower end and when the correlation is rather weak; something that may well be regarded an artifact of using such method.

By applying the method to the LGM tropical temperature model ensemble used in Schmidt et al. (2014), which included 14 models from the PMIP2 and PMIP3 generations, we estimate the climate sensitivity to be 2.6 K (0.7–4.8, 5–95 percentiles). Similarly, applying the method to the mPWP tropical temperature data set of Hargreaves and Annan (2016) gives a climate sensitivity of 2.4 K (0.5–5.0), but with the more uncertain ensemble of models which contributed to PlioMIP1.

With the new generation of climate models, the LGM and mPWP analyses have been widened by the addition of several CMIP6 model outputs. By adding the PMIP4 LGM simulations, we computed a 5–95% interval for climate sensitivity of 0.6–5.2 K. We performed the same analysis by combining PlioMIP1 and PlioMIP2 models and obtained a 5–95% interval of 0.5–4.4 K. However, these results come with some caveats attached. In particular, combining the two model generations of the mPWP could lead to biased results, since the experimental protocol substantially changed in PlioMIP2. An alternative approach is to consider solely the latest version of each model. By doing this we reduce expected redundancy in the ensemble, and so improve our confidence in the result, despite the smaller ensemble sizes. This leads to similarly constrained climate sensitivity of 2.7 (0.6–5.2, 5–95%) for the LGM simulations, and 2.3 (0.4–4.5, 5–95%) for the mPWP simulations. Although most of the computed ranges are wider than the ranges obtained with both OLS or Kalman filtering, the Bayesian framework avoids the underlying assumptions of both methods and in particular, makes us regard the Kalman filtering approach in its current form as too restrictive for the question of emergent constraints.

Nevertheless, our results obtained by analysing the LGM or the mPWP in isolation are broadly consistent with results obtained by other statistical methods used in previous studies. The differences between the way the information is obtained from the paleo record for the mPWP and the LGM and the different dominant climate features of the intervals suggest it may be reasonable to consider these estimates to be statistically independent, given climate sensitivity. It is then possible to combine them within the same Bayesian framework to compute a narrower range of climate sensitivity. By doing so, we evaluated the climate sensitivity to be 2.5 K (0.8–4.0, 5–95%). However, this approach requires independence between the different combined emergent constraints.

It is, in principle, straightforward to include other independent emergent constraints into our Bayesian framework. As well as evidence from historical or present day analyses, other past climates are starting to be explored by modellers and may be potential candidates for future analyses, such as the Eocene, the Miocene and the last deglaciation. Over the next couple of years we expect new outputs for models from CMIP6 and new data analyses to become available, which will enable these preliminary analyses to be compared with results from expanded LGM and mPWP ensembles and improved data estimates.

*Code and data availability.* The Python codes used for the different statistical methods are available from the Bolin Centre Code Repository at https://git.bolin.su.se/bolin/renoult-2020 (https://doi.org/10.5281/zenodo.3928315). The data of the PMIP2 models can be obtained by asking the corresponding modelling groups. The data of the PMIP3 and CMIP6 models can be downloaded from the ESGF Portal at CEDA, located at https://esgf-index1.ceda.ac.uk/. The data of the PlioMIP1 models can be downloaded from Redmine at the School of Earth and Environment of the University of Leeds, located at https://www.see.leeds.ac.uk/redmine/public/. For username and password, email Alan Haywood (a.m.Haywood@leeds.ac.uk). The PRISM3 SST reconstruction can be downloaded from the PRISM/PlioMIP web page, located under "Experiment 1 AGCM version 1.0, Preferred Data" at http://geology.er.usgs.gov/egpsc/prism/prism_1.23/prism_pliomip_data.html, files PRISM3_SST_v1.1.nc and PRISM3_modern_SST.nc. The LGM SAT geological reconstruction can be downloaded from the Supplementary material of Annan and Hargreaves (2013), currently located at http://www.clim-past.net/9/367/2013/cp-9-367-2013-supplement.zip. At time of publication, the data of AWI-ESM-1-1-LR, INM-CM4-8, MIROC-ES2L and MPI-ESM1.2-LR for the LGM and CESM2, EC-EARTH3.3, GISS-E2-1-G, IPSL-CM6A-LR and NorESM1-F for the mPWP are available on ESGF.

*Author contributions.* The BLR method was conceived by JDA and JCH. TM put the project together. The code for the Bayesian framework and for the OLS was written by MR. The code for the Kalman Filter was written by JDA and translated to Python by MR. The code for the conjugate prior approach was written by MR. The statistical analysis were performed by MR. The climate sensitivities of the CMIP6 models were computed by CF. The manuscript was written by MR, JDA, JCH, NS and TM. RO provided the LGM outputs of MIROC-ES2L. UM and MLK provided the LGM outputs of MPI-ESM1.2-LR. GL and XS provided the LGM outputs of AWI-ESM-1-1-LR. QZ and QL provided the mPWP outputs of EC-EARTH3.3.

*Competing interests.* The authors declare that they have no conflict of interest.

*Acknowledgements.* We thank the two anonymous reviewers for their insightful comments. We are very grateful for the extensive work of the scientists involved in the different PMIPs and their efforts to publish and share their data with us. We acknowledge the Bolin Centre for Climate Research at Stockholm University for giving us access to the code repository which allows to freely share our code. We acknowledge the STFC Centre for Environmental Data Analysis (CEDA) in collaboration with the InfraStructure for the European Network for Earth System Modelling (IS-ENES) and the Natural Environment Research Council via the National Centre for Atmospheric Science (NCAS) for making available on the ESGF portal the CMIP5 and CMIP6 dataset. We acknowledge Alan Haywood and Richard Rigby for giving us access to the PlioMIP1 dataset. We acknowledge the Statistical Research Group at the Department of Mathematics at Stockholm University and its director Jan-Olov Persson for his useful comments. Rumi Ohgaito acknowledges support from the Integrated Research Program for Advancing Climate Models (TOUGOU programme) from the Ministry of Education, Culture, Sports, Science and Technology (MEXT), Japan. The simulations using MIROC models were conducted on the Earth Simulator of JAMSTEC. Gerrit Lohmann and Xiaoxu Shi acknowledge support from the PACMEDY project of the Belmont Forum and the PalMod project through BMBF. The simulations using AWI-ESM-1-1-LR were conducted on the German Climate Computing Centre (DKRZ). Qiong Zhang acknowledges the support from Swedish Research Council VR grants 2013-06476 and 2017-04232. The MPI-M contribution was supported by the German Federal Ministry of Education and Research (BMBF) as a Research for Sustainability initiative (FONA) through the project PalMod (FKZ: 01LP1504C). This result is part of the highECS project that has received funding from the European Research Council (ERC) (Grant agreement No.770765) and the CONSTRAIN project, the European Union's Horizon 2020 research and innovation program (Grant agreement No.820829). The analysis and storage of data were performed on resources provided by the Swedish National Infrastructure for Computing (SNIC) at the National Centre at Linköping University (NSC).

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

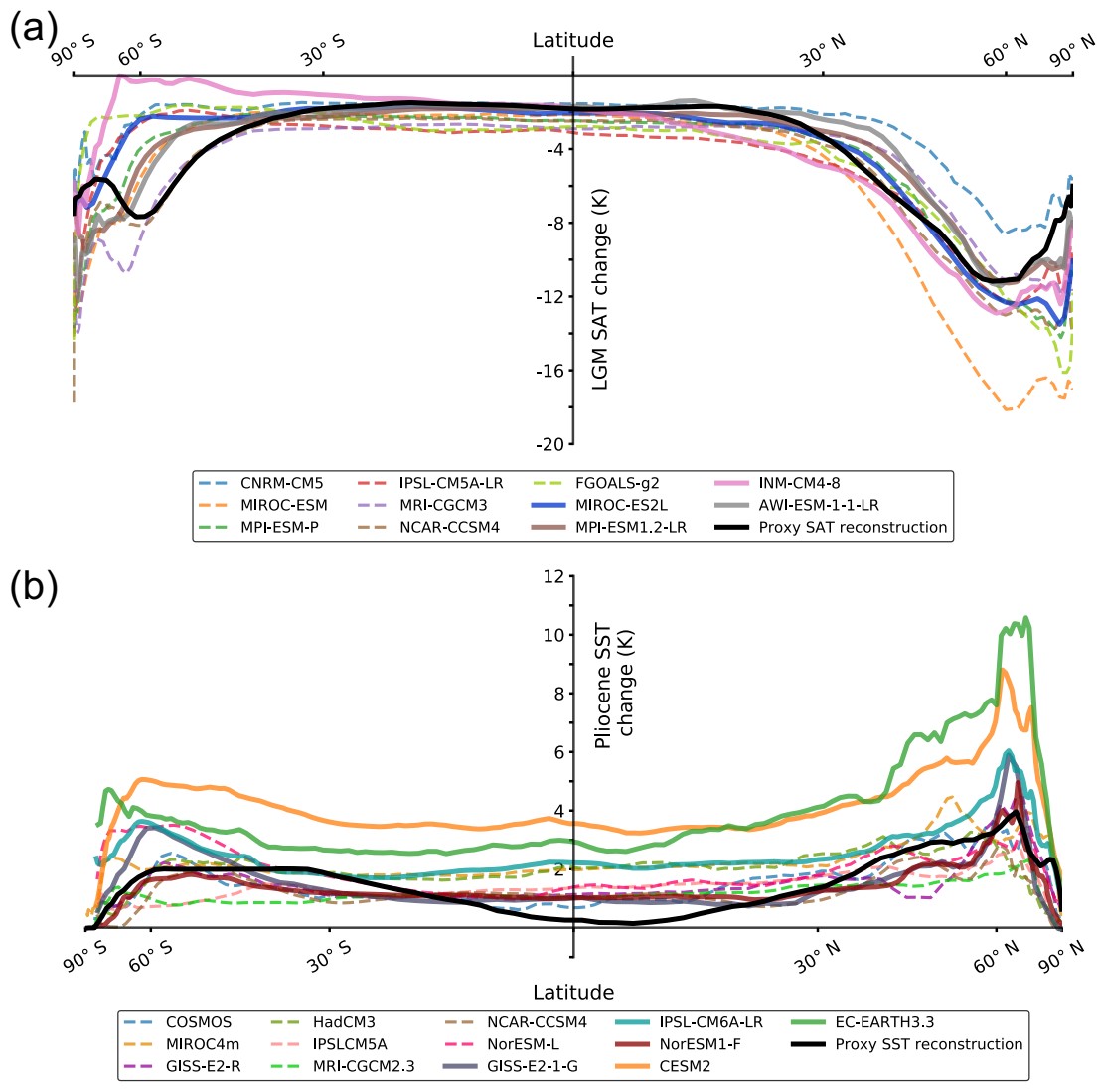

**Figure 1.** Latitudinal distribution of temperature changes relative to pre-industrial for both simulated climates for various climate models and a proxy reconstruction. Dashed lines are models of the CMIP5 generation, while solid lines are models from the CMIP6 generation. All model distributions correspond to 100-year zonal averages when possible; certain CMIP5 PlioMIP1 models were averaged over 30 years. (a), SAT change of the LGM. The solid black line is a multi-proxy ensemble reconstruction taken from Annan and Hargreaves (2013). (b), SST change of the mPWP. The solid black line is the multi-proxy ensemble reconstruction PRISM3 described by Dowsett et al. (2009).

**Table 1.** Models, tropical temperature ($T_{\text{tropical}}$) outputs and Climate Sensitivity ($S$) used in this study

| Experiment | Figure reference | Model | $T^{*}_{\text{tropical}}$ | $S$ | $S$ Reference |
|---|---|---|---|---|---|
| PMIP2 LGM | 1 | MIROC | -2.75 | 4.0 | K-1 Model Developers (2004) |
| PMIP2 LGM | 2 | IPSL | -2.83 | 4.4 | Randall et al. (2007) |
| PMIP2 LGM | 3 | CCSM | -2.12 | 2.7 | Randall et al. (2007) |
| PMIP2 LGM | 4 | ECHAM | -3.16 | 3.4 | Randall et al. (2007) |
| PMIP2 LGM | 5 | FGOALS | -2.36 | 2.3 | Randall et al. (2007) |
| PMIP2 LGM | 6 | HadCM3** | -2.77 | 3.3 | Randall et al. (2007) |
| PMIP2 LGM | 7 | ECBILT** | -1.34 | 1.8 | Goosse et al. (2005) |
| PMIP3/CMIP5 LGM | 8 | CCSM4** | -2.6 | 3.2 | Andrews et al. (2012) |
| PMIP3/CMIP5 LGM | 9 | IPSL-CM5A-LR** | -3.38 | 4.13 | Andrews et al. (2012) |
| PMIP3/CMIP5 LGM | 10 | MIROC-ESM | -2.52 | 4.67 | Sueyoshi et al. (2013) |
| PMIP3/CMIP5 LGM | 11 | MPI-ESM-P | -2.56 | 3.45 | Andrews et al. (2012) |
| PMIP3/CMIP5 LGM | 12 | CNRM-CM5** | -1.67 | 3.25 | Andrews et al. (2012) |
| PMIP3/CMIP5 LGM | 13 | MRI-CGCM3** | -2.82 | 2.6 | Andrews et al. (2012) |
| PMIP3/CMIP5 LGM | 14 | FGOALS-g2** | -3.15 | 3.37 | Yoshimori, pers. comm.[1] |
| PMIP4/CMIP6 LGM | 24 | MPI-ESM1.2-LR** | -2.06 | 3.01 | Mauritsen et al. (2019) |
| PMIP4/CMIP6 LGM | 25 | MIROC-ES2L** | -2.23 | 2.66 | Hajima et al. (2020); Ohgaito et al. (2020) |
| PMIP4/CMIP6 LGM | 26 | INM-CM4-8** | -2.43 | 1.81 | This study |
| PMIP4/CMIP6 LGM | 27 | AWI-ESM-1-1-LR** | -1.75 | 3.61 | This study |
| PMIP3/CMIP5 PlioMIP1 | 15 | CCSM4** | 1.03 | 3.2 | Haywood et al. (2013) |
| PMIP3/CMIP5 PlioMIP1 | 16 | IPSLCM5A | 1.33 | 3.4 | Haywood et al. (2013) |
| PMIP3/CMIP5 PlioMIP1 | 17 | MIROC4m** | 1.99 | 4.05 | Haywood et al. (2013) |
| PMIP3/CMIP5 PlioMIP1 | 18 | GISS ModelE2-R | 1.16 | 2.8 | Haywood et al. (2013) |
| PMIP3/CMIP5 PlioMIP1 | 19 | COSMOS** | 2.18 | 4.1 | Haywood et al. (2013) |
| PMIP3/CMIP5 PlioMIP1 | 20 | MRI-CGCM2.3** | 1.15 | 3.2 | Haywood et al. (2013) |
| PMIP3/CMIP5 PlioMIP1 | 21 | HadCM3** | 1.93 | 3.3 | Randall et al. (2007) |
| PMIP3/CMIP5 PlioMIP1 | 22 | NorESM-L | 1.45 | 2.1 | Haywood et al. (2013) |
| PMIP3/CMIP5 PlioMIP1 | 23 | FGOALS-g2** | 2.14 | 3.37 | Yoshimori, pers. comm.[1] |
| PMIP4/CMIP6 PlioMIP2 | 28 | GISS-E2-1-G** | 0.92 | 2.6 | This study |
| PMIP4/CMIP6 PlioMIP2 | 29 | IPSL-CM6A-LR** | 2.12 | 4.5 | This study |
| PMIP4/CMIP6 PlioMIP2 | 30 | NorESM1-F** | 1.37 | 2.29 | Guo et al. (2019) |
| PMIP4/CMIP6 PlioMIP2 | 31 | CESM2** | 3.5 | 5.3 | Gettelman et al. (2019) |
| PMIP4/CMIP6 PlioMIP2 | 32 | EC-EARTH3.3** | 2.94 | 4.3 | Wyser et al. (2019) |

*For the LGM simulations (generations PMIP2, PMIP3 and PMIP4), the tropical average was defined between $20^{\circ}$ S and $30^{\circ}$ N (Hargreaves et al., 2012). For the mPWP simulations (generations PlioMIP1 and PlioMIP2), the tropical average was defined between $30^{\circ}$ S and $30^{\circ}$ N (Hargreaves and Annan, 2016). All temperature values are defined as changes compared to pre-industrial. **Latest version of a model that was kept for the approach described in Section 3.3.

[1] Calculated using the Gregory method on 150 years of output making it consistent with the values of Andrews et al. (2012).

**Table 2.** Summary of the methods and computed posterior sensitivities.

| Experiment | Method* | 5–95% prior (K) | 5–95% $T^o_{\text{tropical}}$ (K) | Median (K) | 5–95% posterior (K) |
|---|---|---|---|---|---|
| LGM PMIP2 | BF Cauchy prior | 0.5–28.7 | -2.9 – -1.5 | 2.7 | 1.0–4.5 |
| LGM PMIP2 | BF Gamma prior | 0.7–9.5 | -2.9 – -1.5 | 2.6 | 1.0–4.5 |
| LGM PMIP2 | BF Uniform prior | 0.5–9.5 | -2.9 – -1.5 | 2.7 | 0.8–5.0 |
| LGM PMIP2 | OLS predicted CS | n/a | -2.9 – -1.5 | 2.8 | 1.0–4.5 |
| LGM PMIP2 | Kalman filter | 1.7–4.5 | -2.9 – -1.5 | 2.9 | 1.8–4.1 |
| LGM PMIP2 | BF $\alpha$ prior mean=-2 | 0.5–28.7 | -2.9 – -1.5 | 2.7 | 1.0–4.4 |
| LGM PMIP2 | BF $\alpha$ prior mean=-0.5 | 0.5–28.7 | -2.9 – -1.5 | 2.7 | 0.9–4.6 |
| LGM PMIP2+PMIP3 | BF Cauchy prior | 0.5–28.7 | -2.9 – -1.5 | 2.6 | 0.7–4.8 |
| LGM PMIP2+PMIP3 | BF Gamma prior | 0.7–9.5 | -2.9 – -1.5 | 2.6 | 0.9–4.8 |
| LGM PMIP2+PMIP3 | BF Uniform prior | 0.5–9.5 | -2.9 – -1.5 | 2.7 | 0.6–5.4 |
| LGM PMIP2+PMIP3 | OLS predicted CS | n/a | -2.9 – -1.5 | 3.0 | 1.4–4.6 |
| LGM PMIP2+PMIP3 | Kalman filter | 2.0–4.5 | -2.9 – -1.5 | 3.2 | 2.2–4.2 |
| LGM PMIP2+PMIP3 | BF $\alpha$ prior mean=-2 | 0.5–28.7 | -2.9 – -1.5 | 2.6 | 0.8–4.7 |
| LGM PMIP2+PMIP3 | BF $\alpha$ prior mean=-0.5 | 0.5–28.7 | -2.9 – -1.5 | 2.6 | 0.7–4.8 |
| LGM PMIP2+PMIP3 | BF Model inadequacy | 0.5–28.7 | -2.9 – -1.5 | 2.8 | 0.5–5.8 |
| LGM PMIP3 | BF Cauchy prior | 0.5–28.7 | -2.9 – -1.5 | 2.8 | 0.7–5.5 |
| LGM PMIP3 | OLS predicted CS | n/a | -2.9 – -1.5 | 3.4 | 1.3–5.6 |
| LGM PMIP2+PMIP3+PMIP4 | BF Cauchy prior | 0.5–28.7 | -2.9 – -1.5 | 2.7 | 0.6–5.2 |
| LGM "Latest" models | BF Cauchy prior | 0.5–28.7 | -2.9 – -1.5 | 2.7 | 0.7–5.2 |
| LGM "Latest" models | BF Model inadequacy | 0.5–28.7 | -2.9 – -1.5 | 2.8 | 0.5–6.3 |
| mPWP PlioMIP1 | BF Cauchy prior | 0.5–28.7 | -0.8 – 2.4 | 2.4 | 0.5–5.0 |
| mPWP PlioMIP1 | BF $\alpha$ prior mean=2 | 0.5–28.7 | -0.8 – 2.4 | 2.4 | 0.5–4.8 |
| mPWP PlioMIP1 | BF $\alpha$ prior mean=0.5 | 0.5–28.7 | -0.8 – 2.4 | 2.4 | 0.5–5.1 |
| mPWP PlioMIP1 | BF Model inadequacy | 0.5–28.7 | -0.8 – 2.4 | 2.5 | 0.5–5.4 |
| mPWP PlioMIP1+PlioMIP2 | BF Cauchy prior | 0.5–28.7 | -0.8 – 2.4 | 2.3 | 0.5–4.4 |
| mPWP "Latest" models | BF Cauchy prior | 0.5–28.7 | -0.8 – 2.4 | 2.3 | 0.4–4.5 |
| mPWP "Latest " models | BF Model inadequacy | 0.5–28.7 | -0.8 – 2.4 | 2.4 | 0.4–5.0 |
| mPWP and LGM, "Latest" models | BF Cauchy prior | 0.5–28.7 | -2.9 – -1.5 | 2.5 | 0.8–4.0 |
| mPWP and LGM, with CMIP6 | BF Cauchy prior | 0.5–28.7 | -2.9 – -1.5 | 2.4 | 0.7–4.1 |

*BF: Bayesian framework. OLS: Predictive range via ordinary least squares. Truncated-at-zero Cauchy prior: peak=2.5, scale=3. Gamma prior: peak=2, scale=2. Uniform prior: bounded 0–10. The "Latest" models ensembles are those created from the most recent versions of each model (see Section 3.3).

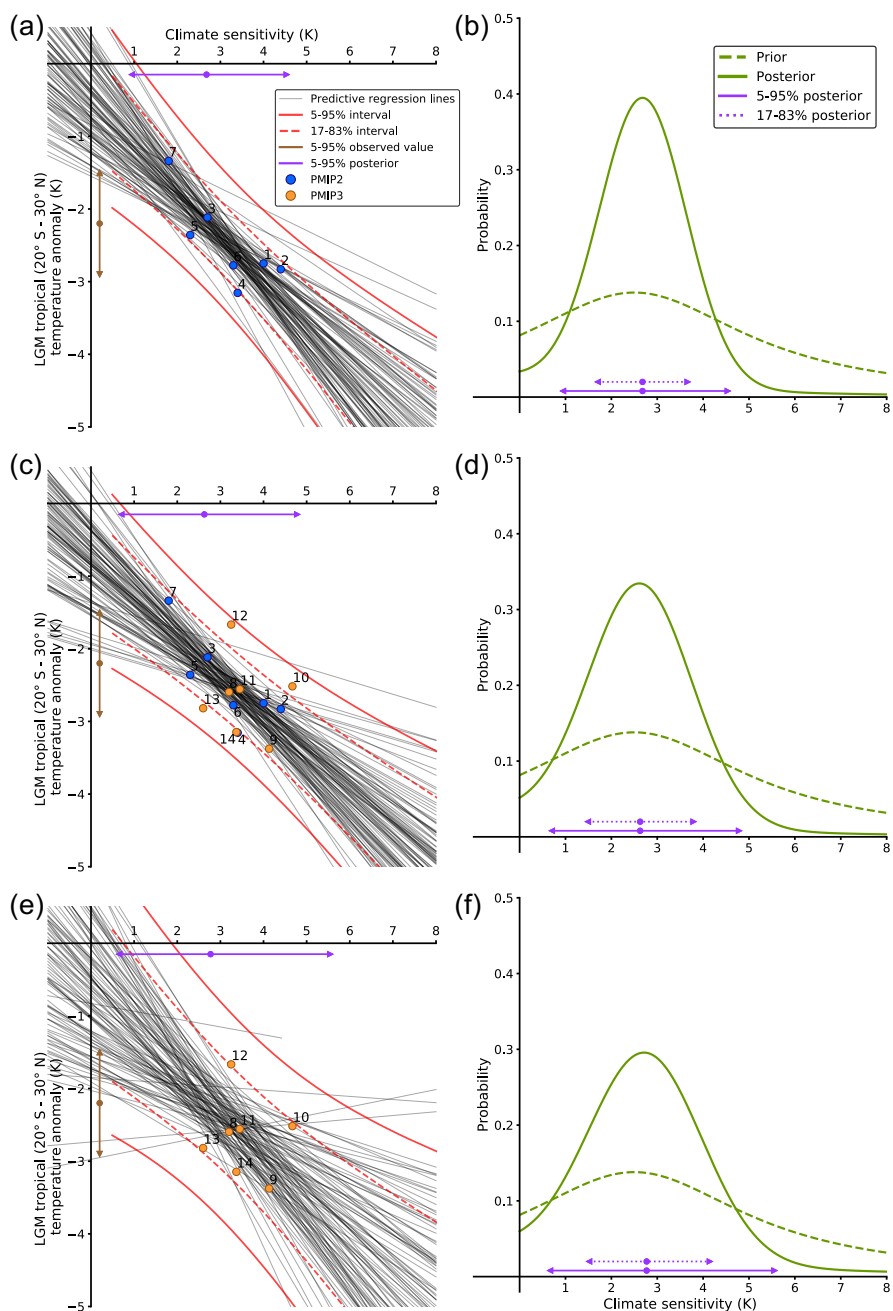

**Figure 2.** LGM northern tropical (20° S–30° N) temperature versus climate sensitivity for the PMIP2 and PMIP3 models. On the left, predictive regression lines sampled with the MCMC method. On the right, corresponding posterior climate sensitivity computed with a Cauchy prior and inferred from a geological reconstruction taken from Hargreaves et al. (2012). (a) and (b), analysis done on the PMIP2 dataset; (c) and (d), analysis done on the PMIP2 and PMIP3 combined dataset; (e) and (f), analysis done on the PMIP3 dataset. The numbers on each point refer to the models used as listed in Table 1.

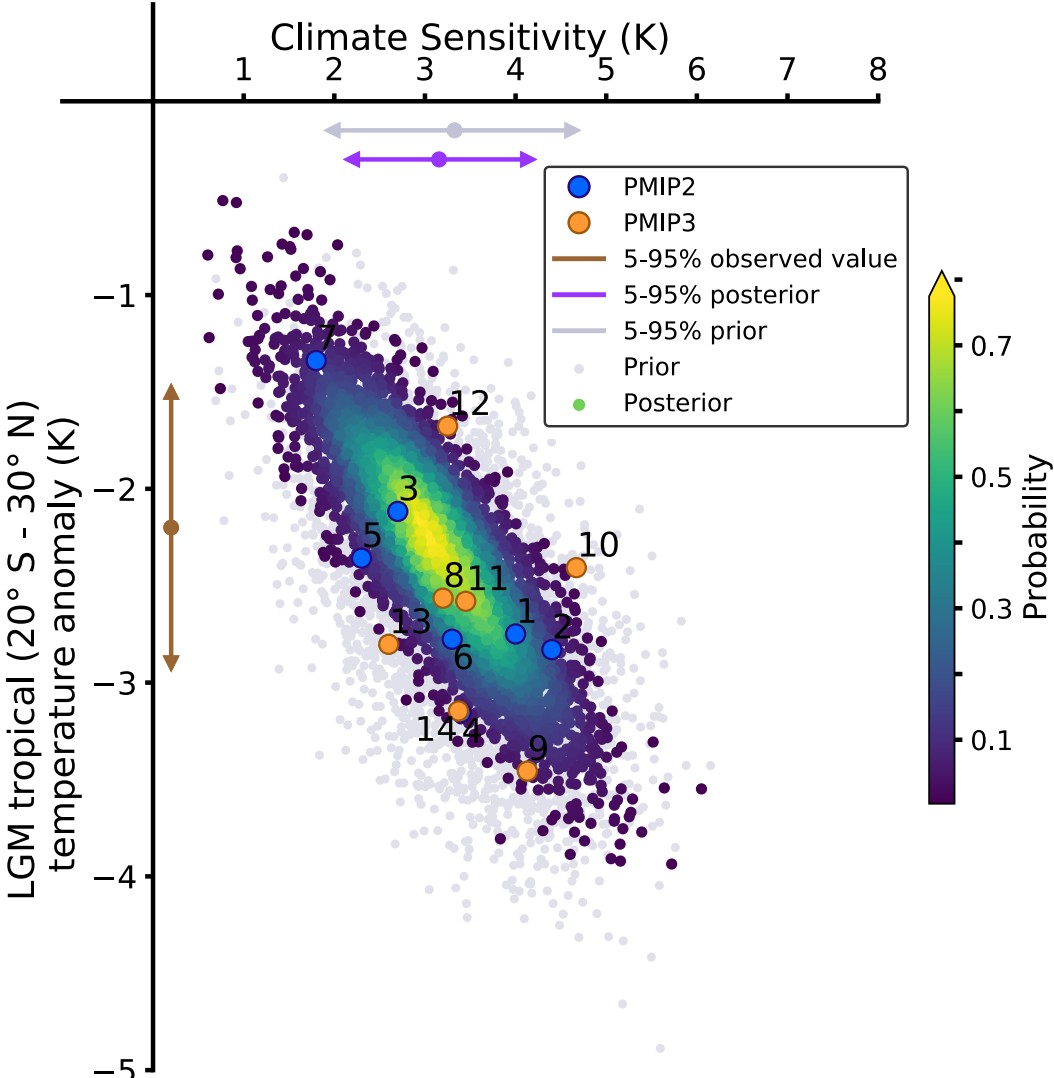

**Figure 3.** LGM northern tropical (20° S–30° N) temperature versus climate sensitivity of the PMIP2 and PMIP3 models. The Kalman filtering is applied on the ensemble of both PMIP2 and PMIP3. The numbers on each point refer to the models used as listed in Table 1.

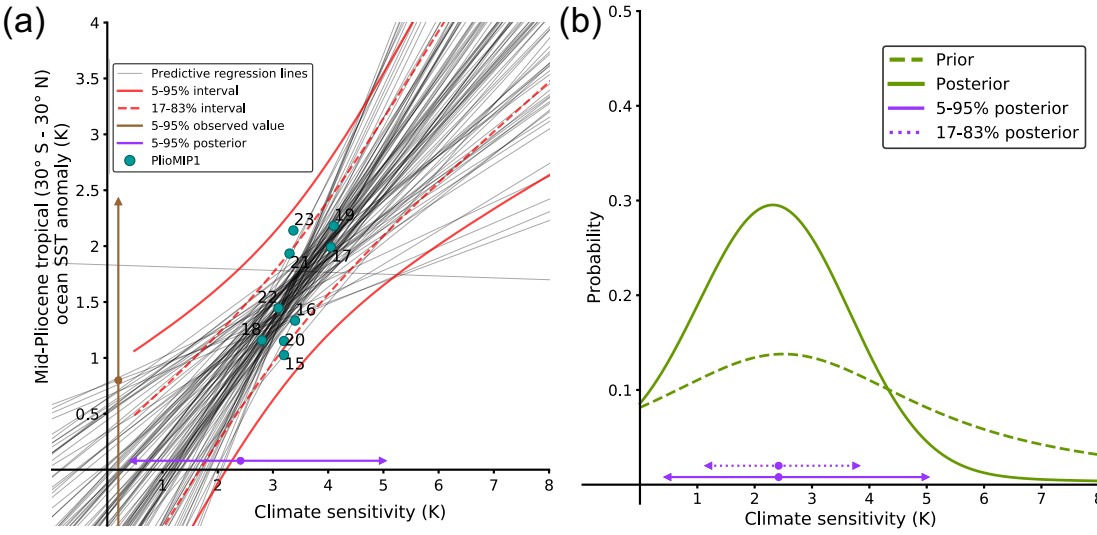

**Figure 4.** mPWP tropical (30° S–30° N) temperature versus climate sensitivity of the PlioMIP1 models. (a), predictive regression lines sampled with a MCMC method. (b), corresponding posterior climate sensitivity computed with a Cauchy prior and inferred from a geological reconstruction taken from Dowsett et al. (2009). The numbers on each point refer to the models used as listed in Table 1.

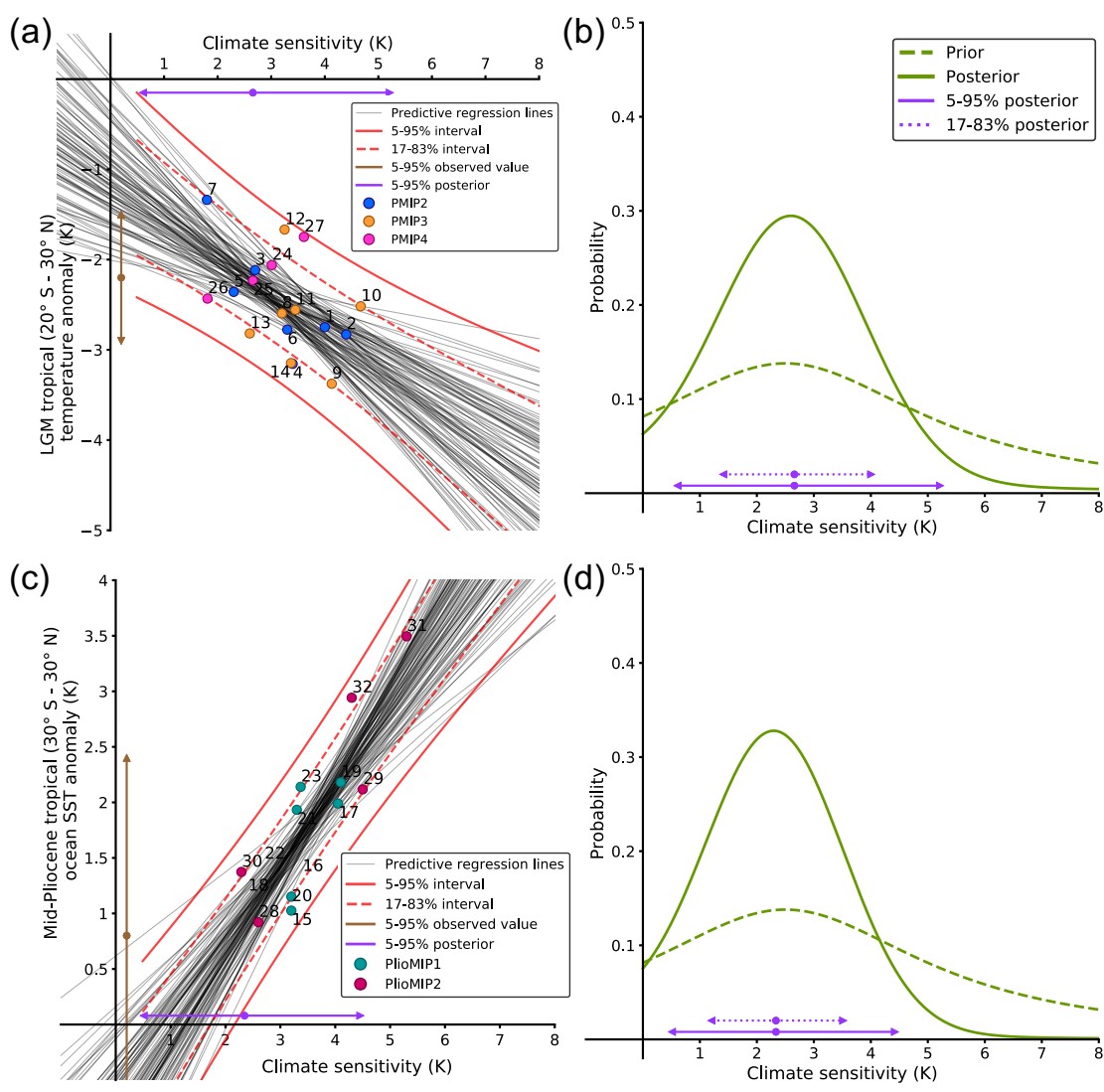

**Figure 5.** Inclusion of the CMIP6 models into the Bayesian method for the LGM and the mPWP. (a), LGM northern tropical (20° S–30° N) temperature versus climate sensitivity of the PMIP2, PMIP3 and PMIP4 models and (b), inferred climate sensitivity. (c), mPWP tropical (30° S–30° N) temperature versus climate sensitivity of the PlioMIP1 and PlioMIP2 models and (d), inferred climate sensitivity. For both inferences, the prior used is a Cauchy distribution defined with a peak of 2.5 and a scale of 3. The numbers on each point refer to the models used as listed in Table 1.

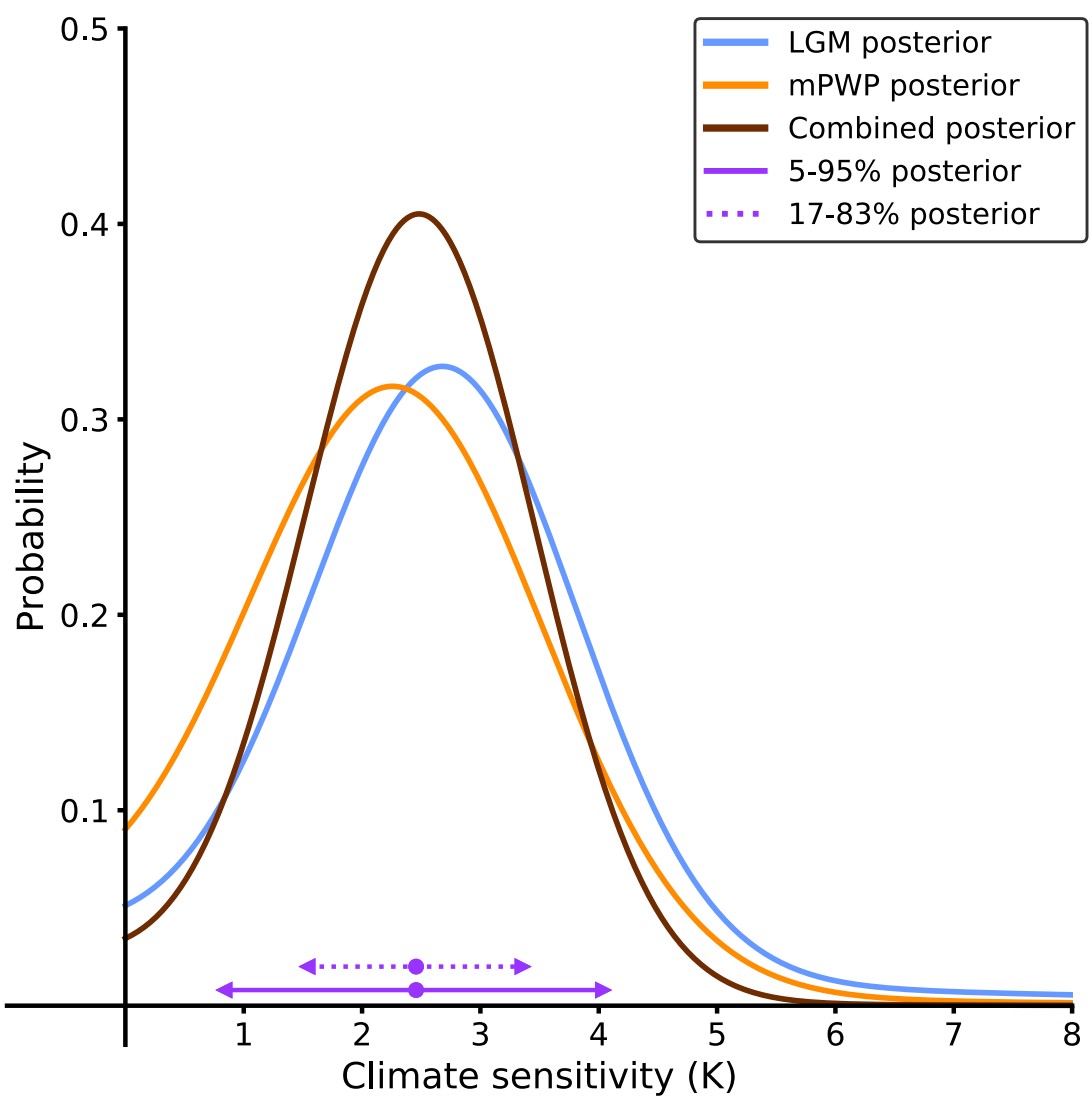

**Figure 6.** Posterior distribution of climate sensitivity computed with a Cauchy prior by combining two assumed independent emergent constraints. The method does not explicitly use both posteriors of the LGM and the mPWP, but use the LGM posterior as the mPWP prior. However, the resulting combined posterior will usually be narrower than the two independent posteriors. For the LGM, the posterior is computed by using the latest model versions of PMIP, including PMIP4. For the mPWP, the posterior is computed by using the latest model versions of PlioMIP, including PlioMIP2.

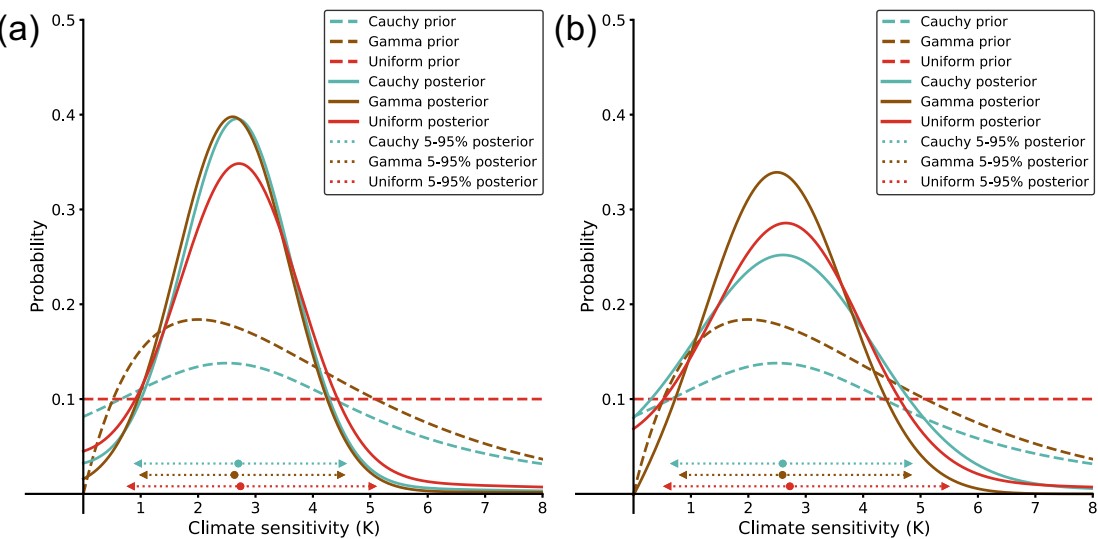

**Figure 7.** Posterior distributions computed with different priors and dataset. (a), posteriors computed with the PMIP2 dataset (strong correlation). (b), posteriors computed with the PMIP2 and PMIP3 dataset combined (weak correlation). The Cauchy prior is defined with a peak of 2.5 and a scale of 3; The Gamma prior is defined with a peak of 2 and a scale of 2; The Uniform prior is bounded between 0 and 10.

## Appendix A: Conjugate priors approach

In section 3, we introduce the NUTS Markov Chain Monte Carlo method, used with the Python package PyMC3 (Salvatier et al., 2016) to compute the posterior distributions of $\alpha$, $\beta$ and $\sigma$ and obtain the likelihood $P(T_{\text{tropical}}|S)$. However, the likelihood model of the Bayesian linear regression is defined as $T_i \sim N(\alpha \times S_i + \beta, \sigma^2)$, where $(T_i, S_i)$ is the $T_{\text{tropical}}$ and $S$ of the $i$ models. Thus, it is possible to choose conjugate priors in this specific case of emergent constraints to avoid using the complex Hamiltonian-based NUTS method. We show here that both approaches lead to similar results.

For the case of the mPWP, we defined the priors $\alpha \sim N(1, 1^2)$, $\beta \sim N(0, 1^2)$ and $\sigma \sim HalfCauchy(Scale = 5)$. Changing the prior $\sigma$ to another family of distribution, such as $\sigma \sim Inverse - Gamma$ leads to a conjugate problem of the form Normal - Inverse Gamma and allows to generate a well-defined form for the posterior distributions of these parameters.

To illustrate this approach, we use the R implementation bayesLMConjugate (where LM stands for Linear Model) of the package spBayes (Finley et al., 2013, 2014). For clarity, a code to perform the MCMC approach is also provided in R, based on the package RSTAN (Stan Development Team, 2019). Conjugate approaches are based on defining priors on the regression parameters conditioned on the (uncertain) error $\sigma$, scaled with a precision matrix $\Lambda_0$. As an example, we define the prior $\sigma \sim InvGamma(a = 0.5, b = 0.5)$, where $a$ represents the shape parameter and $b$ the rate parameter. The prior matrix of the regression parameters (both intercept and slope) is then $N(\mu_0, \sigma^2 \times \Lambda_0)$, where

$$\mu_0 = \begin{matrix} 0 \\ 1 \end{matrix}, \text{the prior mean}$$

and

$$\Lambda_0 = \begin{matrix} 0.5 & 0 \\ 0 & 0.5 \end{matrix}, \text{the prior precision.}$$

The priors provided on the regression parameters in the MCMC method then need to be modified for comparison with the conjugate approach, i.e. conditioned on $\sigma$ itself, which is straightforward to do thanks to the flexibility of PyMC3. Running the code is significantly faster than the use of a MCMC method, and both posterior outputs are compared in Fig. A1. The difference between both methods is minimal and is within the range of precision of MCMC. If we take the full ensemble of PlioMIP1 and PlioMIP2 models simulating the mPWP, and using the posterior distributions of $\alpha$, $\beta$ and $\sigma$ from the conjugate prior method, we estimate a 95% $S$ of 2.3 K (0.5–4.4), compared to a similar value obtained via NUTS. An interesting aspect shown here is that the computed posterior range for $S$ is similar to the one computed with Cauchy and Gamma prior, giving us confidence on the reduced influence of prior distribution in well-correlated and large enough ensemble of data.

Although the choice of conjugate priors would simplify the computation, NUTS (or MCMC methods in general) have the advantage of allowing an explicit and flexible choice of priors for the users. Having such flexibility is a vital element for the analysis presented in this paper. The example taken here to illustrate the Bayesian framework, i.e. the relationship of $T_{\text{tropical}}$ and $S$ is a simple linear regression problem. However, we stipulate that such framework could be used in more complex cases, such as higher complexity emergent constraints relationships, where the use of MCMC methods would become essential.

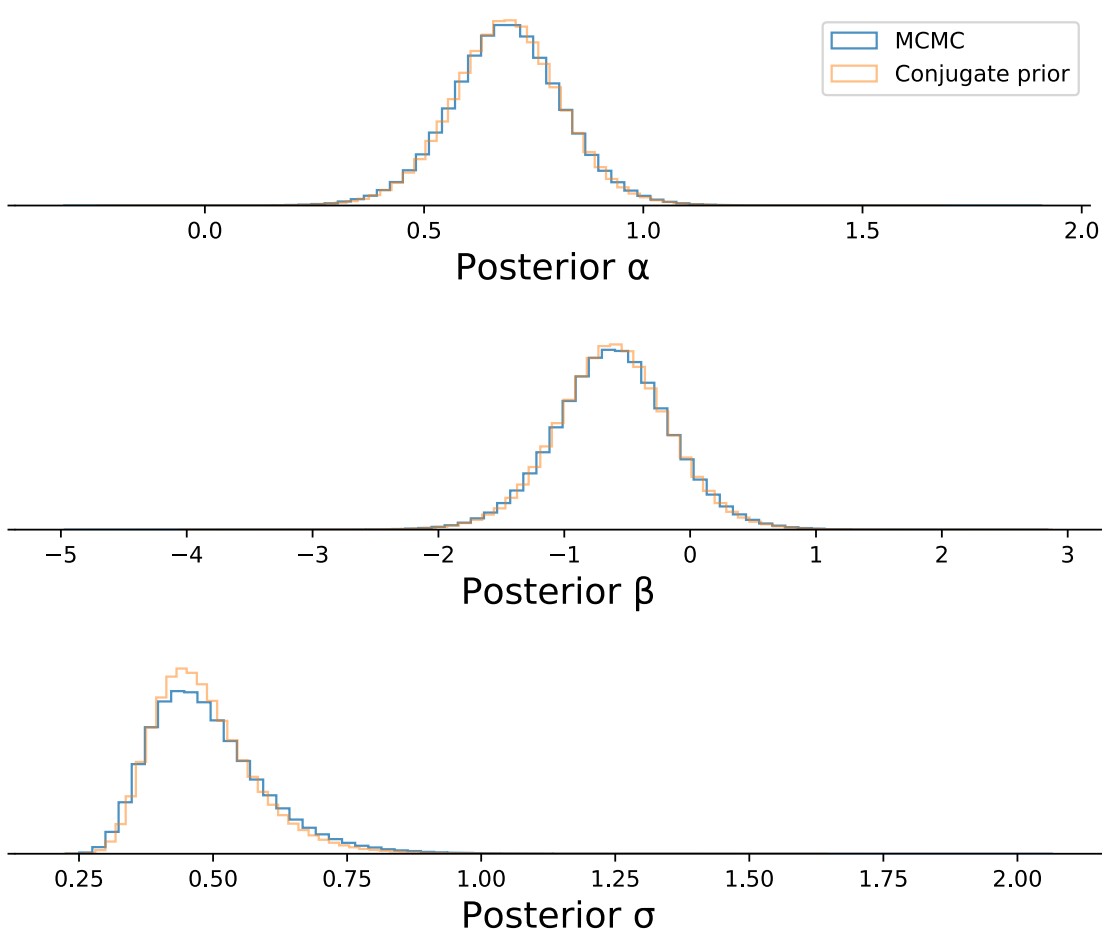

**Figure A1.** Posterior distributions of the three parameters $\alpha$, $\beta$ and $\sigma$ for the case of the combined PlioMIP1 and PlioMIP2 simulating the mPWP. A re-sample of 2 chains of the MCMC method NUTS (in blue) is compared to the conjugate prior approach (in orange). With similar prior distributions on $\alpha$, $\beta$ and $\sigma$, the differences between both methods are minimal.