# Peer review of "A Bayesian framework for emergent constraints: case studies of climate sensitivity with PMIP"

_Climate of the Past, 2019_

## Short Comment (SC1) · 30 Jan 2020

The paper's criticism of the Kalman filter method (section 2.3), as implying – very likely unrealistically – that the model ensemble is a credible predictor before consideration of the observational constraint, almost ruling out posterior estimates outside the model range, is valid and in my view sufficiently important to warrant mentioning in the Abstract.

However, a major weakness of the paper is that it fails to investigate, or even acknowledge the existence of, an objective Bayesian method that has been applied for a very similar purpose, or of the frequentist likelihood ratio method that has also been so applied (Lewis and Grunwald 2018). Objective Bayesian methods use a 'noninformative

prior' that reflects how the expected informativeness of the data about the parameter(s), derived from the likelihood function, varies over the parameter space, and where not all parameters are of interest may also reflect the targeted parameter(s). There is a huge statistical literature on objective Bayesian methods, as there also is on likelihood ratio methods.

Both the aforementioned objective Bayesian and likelihood ratio methods generate uncertainty distributions and ranges that that have been shown, in a perfect model test, to be well calibrated for combining, as well as evaluating separately, independent evidence (Lewis 2018). That is, the uncertainty ranges output by these two methods, although different in statistical nature, are both close to exact confidence intervals. Accordingly, in the long run probabilistic conclusions by an investigator employing either of these methods will on average be true statements, which is surely highly desirable for scientific investigations. That is not in general the case for subjective Bayesian methods (Fraser 2011, Lewis 2014).

Moreover, Bayesian updating does not in general produce satisfactorily calibrated inference when combining evidence, even if the related Bayesian inference from the separate pieces of evidence is well calibrated (Lewis 2013, Lewis 2018). Nor is Bayesian updating satisfactory as a method of incorporating probabilistic prior information, which can however be incorporated under the aforementioned objective Bayesian method. The appropriate way to do so is by treating the prior information not as a prior density to be used in Bayesian updating, but as equivalent to a notional observation with a certain probability density, from which a posterior density has been calculated using Bayes' theorem with a noninformative prior (Hartigan 1965).

In order to achieve satisfactory inference about climate sensitivity when combining evidence, climate scientists need to move on from fundamentally flawed subjective Bayesian methods, and to cease ignoring the existence of objective Bayesian and frequentist (profile) likelihood ratio based methods that are both demonstrably superior.

Nicholas Lewis

References

Fraser, D.A., 2011. Is Bayes posterior just quick and dirty confidence?. Statistical Science, 26(3), pp.299-316.

Hartigan, J.A., 1965: The Asymptotically Unbiased Prior Distribution, Ann. Math. Statist., 36, 4, 1137-1152

Lewis N (2013) Modification of Bayesian updating where continuous parameters have differing relationships with new and existing data. arXiv:1308.2791 [stat.ME]

Lewis N (2014) Objective inference for climate parameters: Bayesian, transformation of variables and profile likelihood approaches. J Clim 27:7270–7284

Lewis N (2018) Combining independent Bayesian posteriors into a confidence distribution, with application to estimating climate sensitivity. J Stat Plan Inference 195, pp.80-92.

Lewis, N. and Grünwald, P. (2018). Objectively combining AR5 instrumental period and paleoclimate climate sensitivity evidence. Climate dynamics, 50(5-6), pp.2199-2216.

---

## Referee Comment (RC1) · Anonymous Referee #1 · 31 Jan 2020

In this paper the authors develop a novel technique to combine emergent constraints. Their main step forward is reconsidering the emergent constraint regression as a likelihood model so that it can be combined with a prior, allowing for Bayesian updating. This is particularly important for estimates of climate sensitivity, whose IPCC range has barely changed since 1990, even though independent lines of evidence have strengthened. The technique is elegant, transparent and I wish I'd come up with it. The accompanying code is also clear. I suggest the authors clarify some of their text and if available include more PMIP4 models.

Minor comments:

11: it's not a 100% clear whether this is a combination of the restricted ensemble of the nonrestricted ensemble. Either clarify, or remove the unrestricted estimate altogether.

[Figure]

16: I don't quite understand the last half of the sentence: "higher bound by construction"

104: I didn't quite understand what "percentage of intervals to contain .." means. Please clarify.

126: typo: roles

139: "observation operator". Operator is unnecessary jargon.

169: A two line explanation of a (one step) Karman filter might benefit readers.

182: Phase 4 of PMIP are used in the study. Please replace explanation by saying not much data is available instead of none.

236-237: I don't think it's necessary to include this test any more.

337: merely → nearly or almost.

291: I'm quite surprised that OLS is more tight. Could you check code or provide an explanation?

349-350: a logical extension of the methodology is to apply it to CMIP, where we find many emergent constraint on the same models. It would be nice if the authors could comment on whether they see this as a problem, given that these models may have similar systematic biases.

374: add 'in a systematic way' or something similar. The principle behind emergent constraints relies on the fact that models deviate from reality, so that's not the problem.

386: pertinent → why not use simpler word such as relevant.

406: ordinary least squares doesn't require capitalization

Fig1 caption: what is a 'wide' ensemble proxy?

Fig2 – Fig9: in the pdf, the colour orange might imply to a tired reader that only PMIP3

is used from the figure on the left. Purple or other dark colour might be more clear. I'm not convinced that all figures are necessary for the paper. The summary in the table may suffice for more regressions, such as the one in Fig 9.

––––––––––––––––––––––––––––

---

## Referee Comment (RC2) · Anonymous Referee #2 · 12 Mar 2020

The paper by Renoult et al presents a new Bayesian method for dealing with emergent constraints for estimating climate sensitivity from palaeoclimate model simulations. I have little expertise in the use of emergent constraints so I will concentrate my comments on the statistical methodology used. For such a simple approach they have made their technique remarkably opaque. For this reason it is hard to recommend an editorial decision for this paper - I will leave it up to the other reviewers to determine novelty and suitability for this journal. However I think there needs to be a considerable improvement in the explanation of the mathematical approaches.

If we start with the OLS method, we have a data set $S_i, T_i$, for $i = 1, \ldots, n$ simulators where we use the model:

$$S_i = \alpha * T_i + \beta + \epsilon$$

[Figure]

And obtain estimates of alpha, beta, and the residual standard deviation sigma. A user comes along and provides us with a new value $T^*$ and we obtain $S^*$ from the fitted model. Uncertainty arises from the potential uncertainties in the estimates of the parameters, and the choice of whether prediction or confidence intervals are used.

So far so good. The authors point out that the Bayesian approach is often superior to these traditional models because of its more sensible handling of uncertainty and the allowance of brining in external information in the form of prior distributions. I agree totally.

Unfortunately here is where things get a little more confusing. The authors then state that the model they want to fit is:

$$p(S|T) = p(T|S)p(S)/p(T),$$

i.e. a standard application of Bayes' theorem which provides us with a posterior distribution of S given T. This is where the notation starts to get into a bit of a mess, because now we're not told where the observations fit in to the model. My guess is that what the authors mean in the above equation (using my notation) is actually:

$$p(S*|T*) = p(T*|S*)p(S*)/p(T*)$$

Where the likelihood p(T*|S*) is actually integrated over the posterior set of parameters from a new linear regression model

$$T_i = \gamma * S_i + \delta + \gamma$$

Where I've named these new slopes/intercepts differently to highlight the different from the previous OLS approach.

This is a more complicated model, and most of has come from guesswork because the authors haven't provided enough information for me to work out exactly what is

happening. I'd really appreciate the authors doing (the quite large job) of either clearing up their maths or making sure that my incorrect assumptions are not made by others.

Apart from this major clanger (on my behalf or theirs), there are a load of other minor points: - The paragraph in the intro which starts "Two recent papers have also addressed..." makes some odd statements about KFs. It points out that everything is Gaussian then states that "it is fairly difficult to generate posterior values which are outside of the prior range". This seems surprising if everything is Gaussian. I haven't read the other paper so perhaps explain more clearly? - The first sentence in the methods section involves, a load of unnecessary commas, which, in my view, makes the sentence, and hence, the definition, of the key concept, of emergent constraints, very hard to, understand. There must be a simpler way of writing it. - Also in that sentence it says '...'then an observation ...'. An observation of what? - L95 should be $N(0, \sigma^2)$ to match standard notation. This mistake is made throughout. There's similarly a bizarre use of $\in fromsettheorytowrite \epsilon \in N(0, \sigma)$ which I think should be $\epsilon \sim N(0, \sigma^2)$ everywhere. - There seems to be a kind of deeper issue that perhaps should be mentioned somewhere that these regression approaches really should involve measurement error (separate from model error as in the Williamson/Samson) paper. The literature on this is well-developed and is pretty easy to include in Bayesian models - L158 the use of sequential updates appears for the first time but I can't really see why this is relevant or used elsewhere? - The Kalman filter method seems like a really important rival approach but despite being given a full subsection 2.3 this only has one paragraph and no mathematical definition. It would be nice to be able to compare the approaches more clearly - PlioMIP appears in L200 without being mentioned before - There really is very little need to use a complex Hamiltonian Monte Carlo method like NUTS on a simple linear regression problem. With a small change in prior from Cauchy to Inv-Gamma the whole problem would become conjugate and could be done exactly on a pocket calculator. - L270 the posterior distributions of what? - L273 and elsewhere. There are some weird mentions about Cauchy distributions having a finite integral whilst Uniform distributions do not. This makes no sense to me (the Uniform is only an improper prior

if it has infinite limits). None of this is referenced so needs clarifying.

---

## Short Comment (SC2) · Comment on Renoult et al. · 19 Mar 2020

Philip Sansom and Daniel Williamson

March 19, 2020

**Summary**

The authors propose an alternative statistical framework for exploiting emergent relationships between multiple climate models in order to constrain future climate. Several advantages are claimed for the proposed method over existing methods, in particular the ability to specify a prior for the quantity of interest directly and independently of the predictive model, and consequently the ability to combine multiple data sources or multiple emergent constraints in a straightforward way. Our view is that the statistical reasoning is difficult to justify for the example of Equilibrium Climate Sensitivity, and impossible for emergent constraints in general. That this is the case is difficult to discern from the current presentation as the model is not written nor justified specifically. This review will attempt to write down the new assumptions and their implications for emergent constraints. The authors make claims regarding the ability of the method to account for model inadequacy, that we argue are not correct. Finally, we attempt to reproduce the results given in the paper and fail for reasons we shall discuss.

**Statistical reasoning for emergent constraints**

When analysing emergent constraints the aim is to obtain a prediction of some quantity of interest $Y^\star$ in the real world, given computer simulations $\mathbf{Y} = (Y_1, \ldots, Y_M)'$ of that quantity, simulations $\mathbf{X} = (X_1, \ldots, X_M)'$ of some other quantity that we can observe in the real world, and an observation $X^\star$ of the real world. Mathematically we aim to form the posterior predictive distribution

$$p(Y^\star \mid X^\star, \mathbf{X}, \mathbf{Y}) = \int p(Y^\star, \boldsymbol{\theta} \mid X^\star, \mathbf{X}, \mathbf{Y}) d\boldsymbol{\theta}.$$

where $\boldsymbol{\theta}$ is a vector of unknown parameters. We omit observation uncertainty for brevity.

Like any mathematical model, a statistical model should be the result of transparent chain of logical reasoning. Unfortunately, the authors do not explicitly specify a probability model, and do not supply the reasoning they use to construct the implied model. This makes it extremely difficult for a non-statistician to judge the claims made for the proposed approach, and difficult even for an experienced analyst to reconstruct the underlying reasoning. We attempt to do this here, as the modelling and assumptions are absolutely critical to the emergent constraints debate and because both this approach and the standard approach cannot be correct at the same time. If there is validity in one of them, users need to have as much access to the assumptions and theory underpinning each in order to work out which.

The standard model specified by Bowman et al. (2018), Williamson & Sansom (2019) and used implicitly in every other emergent constraints study is

$$p(Y^\star, \boldsymbol{\theta} \mid X^\star, \mathbf{X}, \mathbf{Y}) \propto p(Y^\star \mid X^\star, \boldsymbol{\theta}) p(\mathbf{Y} \mid \mathbf{X}, \boldsymbol{\theta}) \pi(\boldsymbol{\theta}).$$

In practice, if $X^\star$ is not known precisely, then a prior $\pi(X^\star)$ must also be specified.

From the manuscript, it is possible to deduce that the implied alternative model is

$$p(Y^\star\boldsymbol{\theta} \mid X^\star, \mathbf{X}, \mathbf{Y}) \propto p(X^\star \mid Y^\star, \boldsymbol{\theta}')\pi(Y^\star)p(\mathbf{X} \mid \mathbf{Y}, \boldsymbol{\theta}')\pi(\boldsymbol{\theta}').$$

The central feature of the proposed approach is to specify $p(X^\star \mid Y^\star)$ and $p(X \mid Y)$ rather than $p(Y^\star \mid X^\star)$ and $p(Y \mid X)$. It is important to realise that these are two fundamentally different statistical models, they cannot both be valid at the same time, and will inevitably result in different inferences for $Y^\star$. Though both equations hold mathematically (as they are valid factorisations of the joint distribution), they imply different conditional independencies in the modelling that need to be physically interpreted and are certainly not interchangable. Therefore, one must consider carefully the underlying reasoning before adopting one or the other. For emergent constraints, $Y^\star$ is usually a measurable property of the future climate and $X^\star$ an observable property of the current or historical climate. Therefore it makes immediate sense to adopt the standard model for emergent constraints, i.e., the future depends upon the past via $p(Y^\star \mid X^\star)$ and $p(Y \mid X)$. In those cases the proposed approach would make no sense because it explicitly states that the past depends on the future via $p(X^\star \mid Y^\star)$ and $p(X \mid Y)$. Equilibrium Climate Sensitivity (ECS) is operationally defined as "the temperature anomaly reached at equilibrium following a instantaneous doubling of $CO_2$". To us, it seems natural to view this as a future climate quantity, and so it makes sense to adopt the standard model for ECS. We also point out that generally statistical reasoning demands that we predict quantities of interest using observables as predictors, though tradition is not as concrete an argument as the causal one.

It might be possible to justify the proposed model for certain quantities of interest, but for an operationally defined future quantity, we would have to be willing to accept the reversal of time's arrow, i.e., past depends on future. It may be that the authors have in mind some alternative definition or interpretation of ECS that renders time's arrow an illusion for this quantity in particular. We are statisticians and take no view on the validity of such a physical argument, but we must strongly insist that the physical argument behind the statistical model be made explicit, well defended, and open to

the scrutiny of other researchers within the field. The models are different and the inferences and conclusions will be different. We feel that transparency is ultimately key to resolving differences.

We note that the given approach would render any claims of causality for emergent constraints impossible, undermining efforts elsewhere in the community to put emergent constraints on a firmer theoretical basis, e.g., Hall et al. (2019).

**Results and reproducibility**

The authors supply all the data used in the study in Table 1, and the observations and priors are given in the text, making it possible to check some of the claims made. We were interested to reproduce these results and compare with our approach. However the statistical model is not stated explicitly in its entirety, nor are the models it is compared to. For the sake of transparency, we interpret the model used by the authors to be

$$
\begin{aligned}
X_m &\sim \text{Normal}(\beta_0 + \beta_1 Y_m, \sigma^2) \quad \text{for } m = 1, \ldots, M \\
X^\star &\sim \text{Normal}(\beta_0 + \beta_1 Y^\star, \sigma^2) \\
Z &\sim \text{Normal}(X^\star, \sigma_Z^2)
\end{aligned}
$$

where $Z$ is the observed value of $X^\star$ and $\sigma_Z^2$ is the observation uncertainty. The default priors are

$$
\begin{aligned}
\beta_0 &\sim \text{Normal}(0, 1^2) \\
\beta_1 &\sim \text{Normal}(-1, 1^2) \\
\sigma &\sim \text{Half-Cauchy}(5) \\
Y^\star &\sim \text{Cauchy}(2.5, 3) \quad \text{truncated at 0.}
\end{aligned}
$$

For the Last Glacial Maximum $z = -2.2$ and $\sigma_Z = 0.7/\Phi^{-1}(0.95) = 0.43$, and for the mid-Pliocene Warm Period $z = 0.8$ and $\sigma_Z = 1.6/\Phi^{-1}(0.95) = 0.97$, where $\Phi^{-1}$ is the quantile function of the normal distribution. This is the model we implement and it is trivial to do so using the Stan probabilistic programming language, however we are unable reproduce many of the reported results. Our comparisons were based on $30\,000$ samples from four independent chains after discarding the first $5\,000$ samples as warm-up for a total of $100\,000$ samples. The chains were checked for convergence both visually and using Gelman-Rubin diagnostics. The medians are usually within believable sampling error, but there are often large discrepancies in the credible intervals.

In particular, for mPWP PlioMIP1 we obtain posterior median $2.5$K ($90\%$ CI $0.4 - 6.3$K) compared to $2.4$K ($90\%$ CI $0.5 - 5.0$K), and for mPWP PlioMIP1+PlioMIP2 we obtain median $2.4$K ($90\%$ CI $0.4 - 5.6$K) compared to $2.4$K ($90\%$ CI $0.4 - 5.0$K). We are, of course, open to the possibility that we have failed to interpret the manuscript text and to implement the correct model, however the Python scripts that accompany the manuscript suggest that we have, and our source code is included for transparency. Examination of those same Python scripts reveals that four chains of only $2\,000$ samples each with no warm-up were used in the production of the reported results (a total of only $8\,000$ samples). Both STAN and PyMC3 (used by the authors) implement the No U-Turn Sampler (NUTS) variant of Hamiltonian Monte Carlo for efficient mixing and fast convergence, so our results should be comparable The lack of warm-up / burn-in period used by the authors is likely to lead to skewed estimates, even using NUTS. Further, the inefficient use of importance sampling to account for the observation uncertainty and the small total number of samples given notorious the difficulty of efficiently sampling from the Cauchy distribution are all likely to contribute to the differences we see when implementing their framework. Unless we have misunderstood their modelling (and by itself this would be an argument for making it explicit in the manuscript), we are not sure that the numbers given in the text actually represent the posteriors of their alternative model faithfully.

Bowman et al. (2018) include explicit expressions for projection using the Kalman filter model (Equations 17 & 23). However using these expressions we are also unable to reproduce the results for the Kalman filter quoted in Table 2 of the manuscript. Examining the Python script that accompanies this manuscript reveals an obvious error in the expression for the posterior variance on Line 80. The authors should therefore revisit the calculation of these credible intervals.

We were able to reproduce the Ordinary Least Squares (OLS) estimates and intervals. However, in Section 2.1 the authors equate OLS with frequentist linear regression. This is incorrect. As discussed in detail in Williamson & Sansom (2019), OLS is a purely algorithmic method of parameter estimation in a mathematical model. The OLS estimates of the mean parameters in a linear regression model are equal to those obtained by frequentist maximum likelihood estimation, *but OLS provides no estimate of uncertainty in either the parameters or the prediction*. Frequentist regression is difficult to justify in a climate change context, but Bracegirdle & Stephenson (2011) presented emergent constraints within a frequentist linear regression framework and this approach has been adopted in many subsequent studies. However, the authors present heuristic uncertainty estimates based on mean-squared errors and on lines 287-288 claim that "OLS" underestimates uncertainty compared to their method. In fact, when standard frequentist regression is used, the inference for ECS in the LGM PMIP2 experiment is very similar to the proposed model with median 2.8K and 90% confidence interval 1.0–4.5K. As Williamson & Sansom (2019) point out, this is equal to an equal tailed 90% Bayesian credible interval under reference priors. Credible intervals from the reference model under the other experiments are less similar to the proposed model, but wider than either the "OLS" or Kalman filter estimates.

[Figure]

**Treatment of model inadequacy**

In the statistical reasoning above we omitted discussion of model inadequacy for brevity. In lines 130–134 and lines 376–378 the authors claim that in their approach model inadequacy can be entirely accounted for by specifying a larger residual variance for reality than the models and it is not necessary to consider differences in the regression parameters. With a minor modification to the proposed model, the intercept can be made independent of the slope, and therefore any additional uncertainty about the intercept in the real world can safely be pushed into the residual since both sources of uncertainty are independent of $Y^\star$. However, any uncertainty in the slope leads to uncertainty about $X^\star$ that is dependent on $Y^\star$, i.e, the width of the predictive interval for $X^\star$ increases with the distance of $Y^\star$ from the prior mean. Therefore any additional uncertainty in the slope in the real world is also conditional on the value of $Y^\star$ and not accounted for by the residual variance. This is a simple matter of geometry and is therefore unavoidable. Williamson and Sansom (2019) developed a coherent elicitation of the regression parameters and structural error that was designed to account for these geometric considerations, and any emergent constraints framework that wishes to account for structural error, whether using the authors approach or the standard one, must grapple with the geometry.

**References**

Bowman, K. W., Cressie, N., Qu, X., & Hall, A. D. (2018). A hierarchical statistical framework for emergent constraints: application to snow-albedo feedback. Geophysical Research Letters, 45(23), 13050–13059. https://doi.org/10.1029/2018GL080082

Bracegirdle, T. J., & Stephenson, D. B. (2012). Higher precision estimates of regional polar warming by ensemble regression of climate model projections. Climate Dynamics, 39(12), 2805–2821. https://doi.org/10.1007/s00382-012-1330-3

all, A., Cox, P., Huntingford, C., & Klein, S. (2019). Progressing emergent constraints on future

climate change. Nature Climate Change, 9(4), 269–278. https://doi.org/10.1038/s41558-019-
0436-6

Williamson, D. B., & Sansom, P. G. (2019). How are emergent constraints quantifying uncer-
tainty and what do they leave behind? Bulletin of the American Meteorological Society, 100,
2571–2588. https://doi.org/10.1175/bams-d-19-0131.1

```
**Load libraries**
library(rstan)

**Model fitting**
fit.blm <- function(X, z, sigmaz,
                    chains = 4, iter = 3e4, warmup = 5e3, cores = 4) {

  ## Fit ``Bayesian Linear Regression'' with truncated Cauchy prior
  data <- list(M = nrow(X), x = X$x, y = X$y, z = z, sigmaz = sigmaz)
  blm  <- sampling(object = blm.model, data = data, chains = chains,
                   iter = iter, warmup = warmup, cores = cores)
  blm  <- as.data.frame(blm)

  ## Return results
  round(quantile(blm$ystar, c(0.50,0.05,0.95)), 1)

}

**STAN code for ``Bayesian Linear Regression''**
blm.code <- "
  data {
    int<lower=0> M;        // number of models
    real x[M];             // predictand
```

```
    real y[M];              // predictor
    real z;                 // observation
    real sigmaz;            // observation uncertainty
  }
  parameters {
    real beta0;             // intercept
    real beta1;             // slope
    real<lower=0> sigma;    // standard deviation
    real xstar;             // latent predictand state
    real<lower=0> ystar;    // latent predictor state
  }
  model {
    // Priors
    beta0 ~ normal( 0.0, 1.0);
    beta1 ~ normal(-1.0, 1.0);
    sigma ~ cauchy( 0.0, 5.0) T[0,];
    ystar ~ cauchy( 2.5, 3.0) T[0,];
    // Models
    for (m in 1:M)
      x[m] ~ normal(beta0 + beta1*y[m], sigma);
    // Observations
    z ~ normal(xstar, sigmaz);
    // Reality
    xstar ~ normal(beta0 + beta1*ystar, sigma);
  }
"

**Compile STAN model**
blm.model = stan_model(model_name = "blm", model_code = blm.code)
```

```
################################
**The Last Glacial Maximum**
################################

**Data**
pmip2 <- data.frame(x = c(-2.70,-2.73,-2.16,-3.18,-2.42,-2.73,-1.37),
                    y = c(+4.00,+4.40,+2.70,+3.40,+2.30,+3.30,+1.80))
pmip3 <- data.frame(x = c(-2.56,-3.46,-2.41,-2.58,-1.68,-2.80,-3.15),
                    y = c(+3.20,+4.13,+4.67,+3.45,+3.25,+2.60,+3.37))
pmip4 <- data.frame(x = c(-2.06,-2.23),
                    y = c(+3.01,+2.66))

**Observations**
z      <- -2.2
sigmaz <-  0.7/qnorm(0.95)

**Results**
pmip2.fit   <- fit.blm(pmip2, z, sigmaz)
pmip3.fit   <- fit.blm(pmip3, z, sigmaz)
pmip23.fit  <- fit.blm(rbind(pmip2,pmip3), z, sigmaz)
pmip234.fit <- fit.blm(rbind(pmip2,pmip3,pmip4), z, sigmaz)

#################################
**The mid-Pliocene Warm Period**
#################################
```

[Figure]

```
**Data**
pliomip1 <- data.frame(x = c(1.03,1.33,1.99,1.16,2.18,1.15,1.93,1.45,2.14),
                       y = c(3.20,3.40,4.05,2.80,4.10,3.20,3.30,2.10,3.37))
pliomip2 <- data.frame(x = c(0.92,2.12,1.37),
                       y = c(2.60,4.50,2.29))

**Observations**
z       <- 0.8
sigmaz <- 1.6/qnorm(0.95)

**Results**
pliomip1.fit  <- fit.blm(pliomip1, z, sigmaz)
pliomip12.fit <- fit.blm(rbind(pliomip1,pliomip2), z, sigmaz)
```

---

## Author Comment (AC2) · 17 Apr 2020

Answer to Anonymous Referee #1

We thank Referee #1 for their careful review of our paper and for the improvements and revisions suggested in their comments. In the following text, we answer to all points discussed by Referee #1, where Referee comments are written as R: and authors comments are written as A:.

R:
In this paper the authors develop a novel technique to combine emergent constraints. Their main step forward is reconsidering the emergent constraint regression as a likelihood model so that it can be combined with a prior, allowing for Bayesian updating. This is particularly important for estimates of climate sensitivity, whose IPCC range has barely changed since 1990, even though independent lines of evidence have strengthened. The technique is elegant, transparent and I wish I'd come up with it. The accompanying code is also clear.
I suggest the authors clarify some of their text and if available include more PMIP4 models.

A:
More PMIP4 models will be added in the revised paper, as several Pliocene and LGM simulations became available during the reviewing process. So far, we obtained 3 more models (with the possible addition of a fourth model).

R:
Minor comments:

A:
Syntax and choice of words will be changed accordingly to the minor comments. Clarifications and change of jargon will be applied when asked to. Specifically, clarifications will be added following these comments:

R: 11: it's not a 100% clear whether this is a combination of the restricted ensemble of the nonrestricted ensemble. Either clarify, or remove the unrestricted estimate altogether.

A: We kept the numbers, but will add the clarification on the use of the restricted ensemble here. We remind here that restricted ensemble refers to an ensemble consisting of the latest version of each model.

16: I don't quite understand the last half of the sentence: "higher bound by construction"

A: Here we meant that the 5th percentile is higher than expected using the OLS method only because of geometrical reasons: If the correlation is weak, then the slope of the regression line is usually reduced and therefore the 5th percentile is higher. This will be made more explicit in this part of the abstract.

We suggest the following correction:

"An interesting implication of this work is that OLS-based emergent constraints on ECS generate tighter uncertainty estimates, in particular at the lower end, suggesting a higher percentile value due to a flatter regression line in case of lack of correlation."

104: I didn't quite understand what "percentage of intervals to contain .." means. Please clarify.

A: We agree this specific statistical jargon can be clarified. What is meant here is that in frequentist confidence intervals, on 100 random 90% confidence intervals arising from separate experiments, 90 intervals would be expected to contain the true interval bounds, which is different than credible intervals, where we believe with 90% probability that the truth lies in the specific interval obtained.

We suggest the following correction:

"The former is the representation of the number of random intervals to contain the true interval bounds (at 90% confidence, this would lead to 90 out of 100 random intervals to contain the true bounds), while Bayesian credible interval is an interval which we believe (with the given probability) to contain the truth."

126: typo: roles

139: "observation operator". Operator is unnecessary jargon.

A: Removed (both).

169: A two line explanation of a (one step) Karman filter might benefit readers.

A: We agree, it will be clarified.

182: Phase 4 of PMIP are used in the study. Please replace explanation by saying not much data is available instead of none.

A: Corrected.

236-237: I don't think it's necessary to include this test any more.

A: We assume Reviewer #1 is referring to the test of other MCMC methods (lines 238-239) which, indeed, can be removed from the study as it is quite trivial.

291: I'm quite surprised that OLS is more tight. Could you check code or provide an explanation?

A: We tried to explain this in the following line (L292 - 293), although it seems to be a limited explanation, which will be therefore extended. The reason behind OLS being tighter is similar to the "higher bound by construction" comment. If the correlation is weak, the OLS will create a regression line that usually has a 90% interval expanding on the range of the ensemble of models. This is not the case for Bayesian regression, as in case of low correlation, the influence will not come from the ensemble of the models but from the prior (here, a Cauchy prior), which has a very wide 90% interval. Therefore, when the correlation is so weak, such as with PMIP3, OLS is much tighter than the Bayesian method. However, it does not imply that either range is closer to reality. They are both different representations of the uncertainties behind each method. The code has been checked and it seems the intervals computed for the OLS method were different than what we sought to represent here. The computations of the intervals were changed to follow the same method as introduced in previous studies (Hargreaves et al., 2012; Schmidt et al., 2014, Hargreaves and Annan, 2016), which is to generate a predictive ensemble aimed at representing the uncertainty in the tropical temperature. This method is actually the one used to generate the 5-95% interval of the Bayesian method. Nevertheless, the "new" intervals are only slightly wider, which leads us to exactly the same interpretation of methods comparison as commented above: Bayesian intervals tend to be wider in case of lack of correlation.

We suggest extending the explanation at lines 292-293 with the following:

"As previously argued for the combination of PMIP2 and PMIP3, the OLS produces a tighter posterior range. In the absence of a correlation, the Bayesian method relaxes to the prior, whereas the OLS method is heavily influenced by the range of the ensemble. However, we emphasise that this does not suggest that either range is closer to reality."

Additionally, the intervals related to the OLS method in Table 1 and in the text will be updated to the new intervals (3 values).

R:

349-350: a logical extension of the methodology is to apply it to CMIP, where we find many emergent constraint on the same models. It would be nice if the authors could comment on whether they see this as a problem, given that these models may have similar systematic biases.

A:
This is an interesting point that will be emphasised in the revised paper. The method was originally designed to be used for different emergent constraints, although we eventually decided to focus on past climates to illustrate the method. There should not be any problem in using CMIP models, as long as the emergent constraints relationship investigated is physically plausible.

337: merely → nearly or almost.

A: Corrected.

374: add 'in a systematic way' or something similar. The principle behind emergent constraints relies on the fact that models deviate from reality, so that's not the problem.

A: Added.

386: pertinent → why not use simpler word such as relevant.

406: ordinary least squares doesn't require capitalization

A: Corrected.

Fig1 caption: what is a 'wide' ensemble proxy?

A: Changed to "multi-proxy ensemble".

R:
Fig2 – Fig9: in the pdf, the colour orange might imply to a tired reader that only PMIP3 is used from the figure on the left. Purple or other dark colour might be more clear. I'm not convinced that all figures are necessary for the paper. The summary in the table may suffice for more regressions, such as the one in Fig 9.

A:
The colours of the figures will be changed accordingly. We agree that not all figures are necessary. Thus, the number of figures will be reduced, as a lot of outputs are summarised in Table 2.

---

## Author Comment (AC3) · 17 Apr 2020

Answer to Anonymous Referee #2

We thank Referee #2 for pointing out sections of the paper that require clarification and providing suggestions and constructive criticism for improvements. In the following text, we answer all the points discussed by Referee #2, where Referee comments are written as R: and authors comments are written as A:.

R: The paper by Renoult et al presents a new Bayesian method for dealing with emergent constraints for estimating climate sensitivity from palaeoclimate model simulations. I have little expertise in the use of emergent constraints so I will concentrate my comments on the statistical methodology used. For such a simple approach they have made their technique remarkably opaque. For this reason it is hard to recommend an editorial decision for this paper - I will leave it up to the other reviewers to determine novelty and suitability for this journal. However I think there needs to be a considerable improvement in the explanation of the mathematical approaches.

If we start with the OLS method, we have a data set Si,Ti, for i = 1,...,n simulators where we use the model:
Si = α ∗ Ti + β + ε
And obtain estimates of alpha, beta, and the residual standard deviation sigma. A user comes along and provides us with a new value T∗ and we obtain S∗ from the fitted model. Uncertainty arises from the potential uncertainties in the estimates of the parameters, and the choice of whether prediction or confidence intervals are used.
So far so good. The authors point out that the Bayesian approach is often superior to these traditional models because of its more sensible handling of uncertainty and the allowance of brining in external information in the form of prior distributions. I agree totally.
Unfortunately here is where things get a little more confusing. The authors then state that the model they want to fit is:
p(S|T ) = p(T |S)p(S)/p(T ),
i.e. a standard application of Bayes' theorem which provides us with a posterior distribution of S given T. This is where the notation starts to get into a bit of a mess, because now we're not told where the observations fit in to the model. My guess is that what the authors mean in the above equation (using my notation) is actually:
p(S ∗ |T ∗) = p(T ∗ |S∗)p(S∗)/p(T ∗)
Where the likelihood p(T*|S*) is actually integrated over the posterior set of parameters
from a new linear regression model
Ti = γ ∗ Si + δ + γ
Where I've named these new slopes/intercepts differently to highlight the different from
the previous OLS approach.
This is a more complicated model, and most of has come from guesswork because the authors haven't provided enough information for me to work out exactly what is happening. I'd really appreciate the authors doing (the quite large job) of either clearing up their maths or making sure that my incorrect assumptions are not made by others.

A: We agree with Reviewer 2, as it led to some confusions also pointed out by Short Comment #2. The description of the model(s) by Reviewer #2 is, however, completely accurate and we thank them for giving us insights on how to clarify our study.
Indeed, the likelihood is built as an integration over the parameters alpha, beta and epsilon. Therefore, the computation of the likelihood, and the updating of the prior, actually calls two different Bayesian processes.

Following the notations of Reviewer 2, the model could be written as:
- The likelihood is a regression model defined by T = γS + δ + ε with the triple of uncertain parameters (γ, δ, ε) which are conditioned on the model ensemble, where T and S are respectively the temperature and sensitivity of a given climate model. Consequently, the likelihood p(T* | S*) for a given S* is an integration over the S* posterior distribution of T* predicted by this regression relation (convolved with observational uncertainty), (γ, δ, ε), conditioned on the model ensemble, where T* represents the observed (geological) value.
- The Bayesian updating of S corresponds then to:
    p(S* | T*) = p(S*) * p(T* | S*)

As accurately described by Reviewer 2, the computation of the likelihood here does not take place within the computation of the emergent constraints model, but forms part of the Bayesian updating, which uses the emergent constraints model.

The notation in the paper will be revised and more clarifications and description of the two stages of the process will be added.

R: The paragraph in the intro which starts "Two recent papers have also addressed. . ." makes some odd statements about KFs. It points out that everything is Gaussian then states that "it is fairly difficult to generate posterior values which are outside of the prior range". This seems surprising if everything is Gaussian. I haven't read the other paper so perhaps explain more clearly?

A: The words we used here were likely too vague. It is true that posterior values could lie outside the prior range. However, it is usually difficult if the observed value is rather uncertain and/or close to the prior mean (which is the case here). We should have rather said that most of the posterior values are generally within the prior range, and this will be clarified in the revised paper.

We suggest the following correction:

"In particular, most of the posterior values would lie in the range covered by the ensemble of models if the observed value is either uncertain and/or close to the prior mean. This is a direct consequence of the joint probability distribution produced by the Kalman filter, which in the case of joint Gaussian distributions, will produce a tighter posterior Gaussian distribution."

R: The first sentence in the methods section involves, a load of unnecessary commas, which, in my view, makes the sentence, and hence, the definition, of the key concept, of emergent constraints, very hard to understand. There must be a simpler way of writing it.

A: This sentence will be clarified to make it easier to understand.

We suggest the following:

"The general method of "emergent constraints" seeks a physically plausible relationship in the climate system between two model variables in an ensemble of results from different climate models. Consequently, an observation of one measurable variable (such as past tropical temperatures) could be used to better constrain the other investigated variable, usually unobserved and difficult to measure (such as climate sensitivity)."

R: Also in that sentence it says '. . .'then an observation . . .'. An observation of what?

A: Here we refer to any kind of measurable quantity in the climate system. For instance, an "observation", or shall we say a measure of the past temperature of the LGM. This will be made more explicit (see suggestion above).

R: L95 should be $N(0, \sigma^2)$ to match standard notation. This mistake is made throughout. There's similarly a bizarre use of $\in$ from set theory to write $\varepsilon \in N(0,\sigma)$ which I think should be $\varepsilon \sim N(0,\sigma^2)$ everywhere.

A: We argue here that $\sigma^2$ could be confusing in some cases, as it implies the user to take the square root to know what $\sigma$ is, which could make it difficult to estimate $2*\sigma$ range, or how many $\sigma$ lies within a certain range. To match standard notation, we will explicitly write the variance as $\sigma^2$, i.e. if $N(m, 0.5^2)$ rather than computing it as $N(m, 0.25)$.
For the use of $\in$, this will be changed to $\sim$.

R: There seems to be a kind of deeper issue that perhaps should be mentioned somewhere that these regression approaches really should involve measurement error (separate from model error as in the Williamson/Samson) paper. The literature on this is well-developed and is pretty easy to include in Bayesian models.

A: This is indeed a relevant point that will be mentioned in the text. We have in fact accounted for observational uncertainty in the natural way in the likelihood, although this was not clearly described in the text.

For the linear regression, it is true that uncertainties exist on S and T of the models, and should, in theory, be included in the Bayesian Linear Regression. However, these uncertainties come from computation methods and are in general very small. For instance, we estimate the measurement error on S to be close to 5% of each value and this is small compared to other uncertainties. In particular, observational errors on S and T from models are small compared to the structural differences that are responsible of the dispersion of the points around the regression line. This point was argued in Hargreaves and Annan (2016) regarding the use of S and T of the mid-Pliocene, where the errors were shown to be small enough to be ignored without a significant impact on the outputs.

Additionally, the strength of the Bayesian Linear Regression is that it does not create a single "best" line, but multiple lines that include all these uncertainties. For instance, it is very likely that multiple lines are already drawn within the range of measurement error of each model. Consequently, we expect the posterior outputs to display variations which could be much less than 0.5 K on both percentiles combined.

R: L158 the use of sequential updates appears for the first time but I can't really see why this is relevant or used elsewhere?

A: Removed "sequentially".

R: The Kalman filter method seems like a really important rival approach but despite being given a full subsection 2.3 this only has one paragraph and no mathematical definition. It would be nice to be able to compare the approaches more clearly

A: The paragraph about the Kalman filter will be extended to give more details, as also requested by Reviewer #1. However, we do not consider the Kalman filter presented here, i.e. a one-step Kalman filter, as an important rival approach, as we state using the climate model ensemble as a prior is in our opinion too strongly restrictive for the question of emergent constraints. We will emphasise this latter point.

R: PlioMIP appears in L200 without being mentioned before

A: Corrected.

R: There really is very little need to use a complex Hamiltonian Monte Carlo method like NUTS on a simple linear regression problem. With a small change in prior from Cauchy to Inv-Gamma the whole problem would become conjugate and could be done exactly on a pocket calculator.

A: There are (at least) three reasons behind the choice of NUTS. The first one is that we consider the Python package PyMC3 explicit and well-described, even for users without statistical background. Thus, using the by-default methods of PyMC3 such as NUTS allow future users to have access to a wide range of online help. The second reason is that the method presented in this study aims at being used in a wider context, i.e. more complex regression problems. The third reason is that using NUTS actually avoids using conjugate priors. We believe that adding more mathematical restrictions on the forms of prior would make the problem less relatable to reality where prior choices may be physically motivated.

R: L270 the posterior distributions of what?

A: Posterior distribution of S (sensitivity). Will be corrected (and other mentions thereafter).

R: L273 and elsewhere. There are some weird mentions about Cauchy distributions having a finite integral whilst Uniform distributions do not. This makes no sense to me (the Uniform is only an improper prior if it has infinite limits). None of this is referenced so needs clarifying.

A: This will be removed as it, indeed, makes little sense in the context.

---

## Author Response (AR3)

Dear Editor,

We now provide a revised version of our manuscript, after some unfortunate delay whom we wish to apologise for. The following takes into accounts the comments given to us during the first and second steps of reviewing. We hope the revisions will bring more clarity to our paper, as requested by the reviewers, and we also managed to expand our ensemble of PMIP4 models, which, we think, makes our paper more suitable for the PMIP4 special issue.

As requested by Reviewer #2, we implemented in our code a conjugate prior approach, illustrated in Appendix A. There was an unfortunate misunderstanding of the approach at first stance, which has been corrected. We thank both the editor and Reviewer #2 for the useful comments in the implementation of this approach. Comparison with the MCMC approach leads to minimal differences which fall within the range of precision of MCMC methods. Thus, the computed posterior S is similar with both methods. For the sake of the user, as we wish to share this method with the broad scientific community, we also added an implementation of the MCMC approach in R.

Additionally, we clarified the comparison between methods based on OLS and our Bayesian approach. There is an obvious difference between previous studies and our research in the choice of S as predicted or predictor. This leads to differences in the computation of posterior S, which are in fact independent of the method chosen. However, we did not explicitly stipulate this aspect in our manuscript (the fact that differences both come from different methods and choice of predictor), and we believe it to be confusing for the reader. This has been corrected in our revised manuscript.

We modified the Bayesian section to make it more explicit that the existence of an emergent constraint relationship is not dependent on the method chosen to analyse it, and that the different analysis methods are expected to draw similar conclusions. However, we do believe that the Bayesian method introduced in this study is the more suitable candidate to exploit the relationship thanks to its transparency and handling of uncertainties.

Finally, we extended the discussion on observational errors. Observational errors are included in our code, in the form of a gaussian uncertainty around the (reconstructed from proxy data) tropical temperature mean. This errors are usually propagation of the different errors carried by the proxy methods used in temperature reconstruction. We acknowledge the existence of other kind of reconstruction which are based on alternative methods to create global temperature field (Bayesian methods, for instance). For the sake of comparison, however, we decided to keep the reconstructions that were used in previous studies, such as Hargreaves et al. (2012) or Hargreaves and Annan (2016).

In the following, the original comments from the reviewers (R) and short comments (C) are written in blue, while our replies (A) are in black.
In addition to our response, we join the two difference files, produced after the first and second minor revisions.

Sincerely,
Martin Renoult

——————————————————————

Reviewer #1

R: In this paper the authors develop a novel technique to combine emergent constraints. Their main step forward is reconsidering the emergent constraint regression as a likelihood model so that it can be combined with a prior, allowing for Bayesian updating. This is particularly important for estimates of climate sensitivity, whose IPCC range has barely changed since 1990, even though independent lines of evidence have strengthened. The technique is elegant, transparent and I wish I'd come up with it. The accompanying code is also clear.
I suggest the authors clarify some of their text and if available include more PMIP4 models.

A: During the revision process, 4 models (INM-CM4-8 and AWI-ESM for the LGM, EC-EARTH3.3 and CESM2 for the mPWP) shared their outputs either on ESGF or directly with us, and therefore have been added to our study. Consequently, numbers, figures and list of authors have been changed. There were some interesting consequences in adding these models, as it reduced the correlation in the case of the LGM and increased it in the case of the mPWP. The overall storyline remains similar as during the first version of this paper. However, some details needed to be changed, in particular regarding the inclusion of PMIP4 / PlioMIP2 with the previous ensembles. These additions made us realise some typos in the Tables and Figures regarding some models: Mostly, some temperatures were erroneous (by only a few decimals, but the correct values were used in the code so there is no impact on the computations) and 2 models were wrongly plotted on the zonal mean plot (Fig. 1). This has been corrected in the revised version.

R:
Minor comments:
11: it's not a 100% clear whether this is a combination of the restricted ensemble of the nonrestricted ensemble. Either clarify, or remove the unrestricted estimate altogether.

A: The words "using the restricted ensemble" have been added, as we considered this estimate rather important in the paper.

R: 16: I don't quite understand the last half of the sentence: "higher bound by construction"

A: The whole sentence has been changed to "An interesting implication of this work is that OLS-based emergent constraints on ECS generate tighter uncertainty estimates, in particular at the lower end, an artifact due to a flatter regression line in case of lack of correlation."

R: 104: I didn't quite understand what "percentage of intervals to contain .." means. Please clarify.

A: Changed to "The former is the representation of the number of random intervals to contain the true interval bounds (at 90% confidence, this would lead to 90 out of 100 random intervals to contain the true bounds), while Bayesian credible interval is an interval which we believe (with the given probability) to contain the truth."

R: 126: typo: roles
139: "observation operator". Operator is unnecessary jargon.

A: Both removed.

R: 169: A two line explanation of a (one step) Karman filter might benefit readers.

A: More details on the one-step process has been written, and the Kalman Filter section has been expanded.

R: 182: Phase 4 of PMIP are used in the study. Please replace explanation by saying not much data is available instead of none.

A: Added.

R: 236-237: I don't think it's necessary to include this test any more.

A: The other MCMC tests have been removed.

R: 291: I'm quite surprised that OLS is more tight. Could you check code or provide an explanation?

A: The following explanation has been added: "As previously argued for the combination of PMIP2 and PMIP3, the OLS produces a tighter posterior range. In the absence of a correlation, the Bayesian method relaxes to the prior, whereas the OLS method is heavily influenced by the range of the ensemble. However, we emphasise that this does not suggest that either range is closer to reality."

Additionally, the number coming from OLS-based approaches have been updated to their predictive intervals values (minor changes which make them more consistent with previous studies).

R: 349-350: a logical extension of the methodology is to apply it to CMIP, where we find many emergent constraint on the same models. It would be nice if the authors could comment on whether they see this as a problem, given that these models may have similar systematic biases.

A: This is an interesting point which could lead to promising research in the future! The following text has been added in "Combining multiple constraints" : "A logical extension of the approach would be to apply it to the ensemble of models of CMIP, where multiple emergent constraints exist for the same models. In theory, this should be possible as long as the investigated relationships are physically plausible. This goes beyond the scope of our study, which uses the paleoclimates as an example for the method, and is left for future research."

R: 337: merely → nearly or almost.

A: Changed to "nearly"

R: 374: add 'in a systematic way' or something similar. The principle behind emergent constraints relies on the fact that models deviate from reality, so that's not the problem.

A: Added.

R: 386: pertinent → why not use simpler word such as relevant.

A: Changed to "relevant".

R: 406: ordinary least squares doesn't require capitalization

A: Changed in the whole manuscript.

R: Fig1 caption: what is a 'wide' ensemble proxy?

A: Changed to "multi-proxy ensemble"

R: Fig2 – Fig9: in the pdf, the colour orange might imply to a tired reader that only PMIP3 is used from the figure on the left. Purple or other dark colour might be more clear. I'm not convinced that all figures are necessary for the paper. The summary in the table may suffice for more regressions, such as the one in Fig 9.

A: The colours were updated in the corresponding figures to have a more distinct differences. Fig. 7 ("Latest version approach") and Fig. 9 ("Model inadequacy") were removed as it is summarised already in Table 1.

————————————————————
Reviewer #2

R: The paper by Renoult et al presents a new Bayesian method for dealing with emergent constraints for estimating climate sensitivity from palaeoclimate model simulations. I have little expertise in the use of emergent constraints so I will concentrate my comments on the statistical methodology used. For such a simple approach they have made their technique remarkably opaque. For this reason it is hard to recommend an editorial decision for this paper - I will leave it up to the other reviewers to determine novelty and suitability for this journal. However I think there needs to be a considerable improvement in the explanation of the mathematical approaches.
If we start with the OLS method, we have a data set Si,Ti, for i = 1,...,n simulators where we use the model:
$S_i = \alpha * T_i + \beta + \varepsilon$

And obtain estimates of alpha, beta, and the residual standard deviation sigma. A user comes along and provides us with a new value $T_*$ and we obtain $S_*$ from the fitted model. Uncertainty arises from the potential uncertainties in the estimates of the parameters, and the choice of whether prediction or confidence intervals are used.

So far so good. The authors point out that the Bayesian approach is often superior to these traditional models because of its more sensible handling of uncertainty and the allowance of brining in external information in the form of prior distributions. I agree totally.

Unfortunately here is where things get a little more confusing. The authors then state that the model they want to fit is:

$p(S|T) = p(T|S)p(S)/p(T)$,

i.e. a standard application of Bayes' theorem which provides us with a posterior distribution of S given T. This is where the notation starts to get into a bit of a mess, because now we're not told where the observations fit in to the model. My guess is that what the authors mean in the above equation (using my notation) is actually:

$p(S_*|T_*) = p(T_*|S_*)p(S_*)/p(T_*)$

Where the likelihood $p(T^*|S^*)$ is actually integrated over the posterior set of parameters from a new linear regression model

$T_i = \gamma * S_i + \delta + \gamma$

Where I've named these new slopes/intercepts differently to highlight the different from the previous OLS approach.

This is a more complicated model, and most of has come from guesswork because the authors haven't provided enough information for me to work out exactly what is happening. I'd really appreciate the authors doing (the quite large job) of either clearing up their maths or making sure that my incorrect assumptions are not made by others.

A: First, as suggested in the last equation, the parameters of the OLS-based approach have been given different names, γ, δ and ζ, to emphasise the differences with the Bayesian framework, and also because their values are, indeed, different from α, β and ε. Besides being different methods, the OLS-based approach (or shall we say, the "forward" approach, if one direction can be given) will lead necessarily to different outputs than the Bayesian approach ("inverse" regression), since the errors will not be taken into account on the same variable. In fact, as written in Short Comment #2, both approaches should not be able to exist at the same time with the same parameters, since one variable (T or S) can not be dependent and independent of the same error parameter. This is the reason why γ, δ are different from α, β, but mainly why ζ is different from ε since they have different physical meanings. In this sense, it should be relatively difficult to compare the OLS approach given in this paper (or in previous studies, as written with S as predicted variable) to the "inverse" Bayesian approach we introduce here from a statistical perspective. However, we note that both methods have the same objective of predicting S by considering a certain ensemble of "information" coming from climate models. That is, the relationship of emergent constraint (between T and S) exists independently of the statistical method that will be used to analyse it. That includes forward/inverse regression but also OLS and Bayesian approaches. Thus, one should not expect critical differences of posterior S no matter what method is chosen, as all methods exploit the same assume relationship but only seek to compute different unknown parameters.

The following text has been added in the revised manuscript:

Choosing S as the predictor (Eq. 3) will cause some differences to the inference of posterior S compared to the OLS-based approach introduced in Eq. 1. The plausibility of the existence of an emergent constraint between S and T_tropical is independent of the method chosen. Whether T_tropical is a predicted or predictor variable, or whether the applied method uses OLS or Bayesian statistics, the methods estimate different unknown parameters to investigate a similar assumed relationship within the model ensemble; and so it is expected that these different methods will yield similar but not identical results. This was previously argued in the context of hierarchical statistical model for emergent constraints by Tingley et al. (2012). The Bayesian approach with S as the predictor is appropriate for emergent constraint analyses thanks to its transparency and handling of uncertainties. This has been explored by Sherwood et al. (2020), and is also investigated in this study. Thus, here we explore the implementation of the Bayesian

method for emergent constraint analyses, to models and data that have already been investigated with alternative methods (Hargreaves et al., 2012; Hargreaves and Annan, 2016).

Having said so, a source of confusion, as pointed by Reviewer #2, were the annotations and the model not being explicit enough. We decided to change the name of the section "Bayesian Linear Regression" to "Bayesian Framework" to emphasise the 2-step Bayesian process we are using in the paper. Specifically, the Bayesian updating process which aims at computing the final posterior S is

$$p(S|T^o\_tropical) = p(T^o\_tropical |S)p(S)/p(T^o\_tropical)$$

The value $T^o\_tropical$ is then a reconstruction from proxy data of the tropical temperature of either the LGM or the mPWP and allows to compute a posterior S in the emergent constraints framework. As noted by Reviewer #2, here, the likelihood $p(T^o\_tropical |S)$ (noted $P(T^* | S^*)$ by Reviewer #2) was an integration over the posterior S of the set of parameters α, β and ε, coming from the second Bayesian process, the Bayesian Linear Regression. This BLR process is constrained on the model ensemble and in particular their values S and T_tropical. Thus, the BLR model is defined as:

$$Ti\_tropical = α * Si + β + ε, ε \sim N(0,σ^2)$$

Where the "i" refers to the index of one of the climate model used. In this way, we create a likelihood (the posterior of the BLR) P(T_tropical | S) by integrating over S, for the Bayesian updating process. We explicitly stated that P(T_tropical | S) is approximated as P(T_tropical = $T^o\_tropical$ | S) through an importance sampling method, allowing us to insert the proxy reconstruction and create a series of weights to update the prior P(S) into the posterior of interest, P(S | $T^o\_tropical$).

We hope this clarification will make the comprehension of our method easier than before. We decided to explicitly split the 2 Bayesian processes, as the BLR is only used to compute the likelihood necessary for the Bayesian updating. The Bayesian updating, in fact, is the only way of computing the climate sensitivity; Our implementation of emergent constraints in a Bayesian framework, however, requires both processes.

R: The paragraph in the intro which starts "Two recent papers have also addressed. . ." makes some odd statements about KFs. It points out that everything is Gaussian then states that "it is fairly difficult to generate posterior values which are outside of the prior range". This seems surprising if everything is Gaussian. I haven't read the other paper so perhaps explain more clearly?

A: It has been changed to: "In particular, most of the posterior values would lie in the range covered by the ensemble of models if the observed value is either uncertain and/or close to the prior mean. This is a direct consequence of the joint probability distribution produced by the Kalman filter, which in the case of joint Gaussian distributions, will produce a tighter posterior Gaussian distribution. Because of that, it does not appear to correspond to the choice which is usually made, albeit implicitly."

R: The first sentence in the methods section involves, a load of unnecessary commas, which, in my view, makes the sentence, and hence, the definition, of the key concept, of emergent constraints, very hard to understand. There must be a simpler way of writing it.

A: Changed to: "The general method of "emergent constraints" seeks a physically plausible relationship in the climate system between two model variables in an ensemble of results from different climate models. Consequently, an observation of one measurable variable (such as past tropical temperatures) could be used to better constrain the other investigated variable, usually unobserved and difficult to measure (such as climate sensitivity)."

Additionally, the following has been added later on in the same paragraph, as we think it is a source of confusion for our approach: "Although the unobserved variable is usually taken as a

future variable, the emergent constraints theory can be used with two variables within the same time frame, as long as the relationship is plausible."

R: Also in that sentence it says '. . .'then an observation . . .'. An observation of what?

A: Changed to "an observation of one measurable variable (such as past tropical temperatures) could be used to better constrain the other investigated variable, usually unobserved and difficult to measure (such as climate sensitivity)."

R: L95 should be $N(0, \sigma^2)$ to match standard notation. This mistake is made throughout. There's similarly a bizarre use of $\in$ from set theory to write $\varepsilon \in N(0,\sigma)$ which I think should be $\varepsilon \sim N(0,\sigma^2)$ everywhere.

A: We matched standard notation by writing $N(0, \sigma^2)$ in the text. We also wrote $\varepsilon \sim N(0,\sigma^2)$ in the text.

R: There seems to be a kind of deeper issue that perhaps should be mentioned somewhere that these regression approaches really should involve measurement error (separate from model error as in the Williamson/Samson) paper. The literature on this is well-developed and is pretty easy to include in Bayesian models.

A: The following text has been added regarding observational errors. In particular, (model) observational errors have been shown to have insignificant impact on the results in the case of this emergent constraint in a previous study on the mPWP (Hargreaves and Annan, 2016). For the case of observational errors on the tropical paleotemperature reconstruction, observational errors are already included in the paper under the form of Gaussian uncertainty. For the case of the LGM, the temperature has been computed as -2.2 +/- 0.4 (1 standard deviation), and for the mPWP, at +0.8 +/- 1 (1 std). For both cases, these values come from global multi-proxy integration which considered error propagation and uncertainties from region with low or no proxy data available. For instance, Annan and Hargreaves (2013) performed multi-linear regression based on the multi-proxy ensemble of MARGO, and using climate models as predictor (by efficiently comparing them and estimating their predicting abilities compared to the proxy data) to estimate temperature change when no proxy data are available. Fortunately for the case of the LGM, it leads to a rather low uncertainty in the tropical region. This is not the case when considering global LGM temperature, where the observational uncertainty is substantially high due to uncertainties regarding the Northern ice sheets and dust loading, mainly. For the mPWP, the observational uncertainty chosen is the widest given in the literature, as we did not have specific reasons for choosing low uncertainties in the tropical reconstruction of the mPWP. However, choosing more constrained observational errors has inherent drawbacks on the inferred sensitivity, as shown in Hargreaves and Annan (2016). In the provided code, the user can freely change the reconstructed temperature mean and standard deviation. Additionally, although we argue that observational errors on model temperature can be neglected since they are small compared to the spread of the ensemble, they can be easily implemented in the code. This would require the user to randomly sample model temperature, considering their uncertainty. This has not been explored in our paper, as we stipulate it has a minimal, if not negligible, impact on the computation of posterior S. Although we acknowledge the existence of other approaches in producing global temperature field for the mPWP, for the sake of comparison, we preferred using the reconstructions that have been used in past studies dealing with similar topics (e.g. Schmidt et al., 2014; Hargreaves and Annan, 2016).

"It is not clear if observational errors have always been adequately accounted for in previous emergent constraints research. Our approach provides a natural framework for this, as the likelihood can include the uncertainty of the observational process as we have done. Recent studies have investigated Bayesian ways of integrating uncertainties on proxy reconstructions into global temperature field (e.g. Tierney et al., 2019). For the sake of comparison with Hargreaves et al. (2012), Schmidt et al. (2014), Hargreaves and Annan (2016), we use the reconstructions and observational errors adopted in these studies which are based on multiple linear regressions and model-proxy cross-validation.
However, we have ignored uncertainties in the calculation of the model values of S and T_tropical as, while they are poorly quantified, we believe them to be too small to materially affect our result.

In fact, it has been argued for the case of the mPWP that observational errors on S and T_tropical are small compared to the structural differences responsible of the dispersion of the points around the regression line and thus can be neglected (Hargreaves and Annan, 2016)."

R: L158 the use of sequential updates appears for the first time but I can't really see why this is relevant or used elsewhere?

A: Removed.

R: The Kalman filter method seems like a really important rival approach but despite being given a full subsection 2.3 this only has one paragraph and no mathematical definition. It would be nice to be able to compare the approaches more clearly

A: The mathematical definitions were added. Additionally, we added more references to the fact that we do not think the Kalman Filter is a relevant rival approach in that particular case, as it is too restrictive on its prior, and consequently, on its posterior S.

R: PlioMIP appears in L200 without being mentioned before

A: Corrected to "Pliocene Modelling Intercomparison Project (PlioMIP)".

R: There really is very little need to use a complex Hamiltonian Monte Carlo method like NUTS on a simple linear regression problem. With a small change in prior from Cauchy to Inv-Gamma the whole problem would become conjugate and could be done exactly on a pocket calculator.

A: Adding more mathematical restrictions on the form of prior would make the problem less relatable to reality where prior choices may be physically motivated. However, in Appendix A, we have illustrated the result obtained by the reviewer's suggestion. If an Inverse Gamma prior is given on the MCMC method, the computed posterior S will be similar than with conjugate prior, considering the precision range of MCMC (two decimals). The numbers obtained with NUTS have also been checked with alternative packages (RSTAN, in R), where other MCMC methods are provided (namely Hamiltonian and Metropolis-Hastings). The main difference we highlight between RSTAN, PyMC3 and the conjugate prior approaches is that RSTAN and PyMC3 are based on defining clearly independent prior standard deviations on the parameters (slope, intercept and sigma) of the regression lines, while the conjugate approaches (packages such as bayesLMconjugate in R, NIG in Python or applying directly the mathematical equations for a normal-inverse gamma conjugate problem) are based on defining a prior variance-covariance matrix and giving an uncertainty represented by $Sigma^2 * V$, which can be more difficult to link to reality. Nevertheless, we showed that applying the exact same prior distributions in RSTAN, PyMC3 and BayesLMConjugate lead to the same outputs (within the range of precision of MCMC). An interesting conclusion of this work is that the choice of Inverse Gamma prior also leads to the same computed S than Cauchy and Gamma prior, which gives us confidence in the reduced effect of prior distributions when the ensemble is large enough and well-correlated.

R: L270 the posterior distributions of what?

A: Posterior distributions of S. This has been corrected in the text.

R: L273 and elsewhere. There are some weird mentions about Cauchy distributions having a finite integral whilst Uniform distributions do not. This makes no sense to me (the Uniform is only an improper prior if it has infinite limits). None of this is referenced so needs clarifying.

A: This has been removed/corrected. We still mention this once, but have explicitly stated that in our case, the prior can not be considered as improper since it is bounded. However, one could indeed choose a non-bounded Uniform prior to infer S, in which case the prior would be improper.

————————————————————
Short Comment #1

No changes were applied in the manuscript in response to SC#1. As we argued in the original answer, we do not think that the Kalman filter should be mentioned in the abstract to avoid an overload of information. Additionally, we did not wish to add other Bayesian approaches as it is beyond the scope of this paper. The initial online response was:

C: The paper's criticism of the Kalman filter method (section 2.3), as implying – very likely unrealistically – that the model ensemble is a credible predictor before consideration of the observational constraint, almost ruling out posterior estimates outside the model range, is valid and in my view sufficiently important to warrant mentioning in the Abstract.

A: This, indeed, is a relevant point. As pointed out by Reviewer #2, the Kalman Filter could appear as a significant rival approach. However, we consider it too restrictive, for the reasons reminded here by the author of the comment. Nevertheless, the main scope of the study is not to criticise the Kalman filter method, but to present a Bayesian method which we think is more appropriate to the question of emergent constraints. Therefore, we do not consider adding references to the Kalman Filter in the abstract, mainly to avoid an overload of information that could mislead the reader.

C: However, a major weakness of the paper is that it fails to investigate, or even acknowledge the existence of, an objective Bayesian method that has been applied for a very similar purpose, or of the frequentist likelihood ratio method that has also been so applied (Lewis and Grunwald 2018). Objective Bayesian methods use a 'noninformative prior' that reflects how the expected informativeness of the data about the parameter(s), derived from the likelihood function, varies over the parameter space, and where not all parameters are of interest may also reflect the targeted parameter(s). There is a huge statistical literature on objective Bayesian methods, as there also is on likelihood ratio methods.
Both the aforementioned objective Bayesian and likelihood ratio methods generate un- certainty distributions and ranges that that have been shown, in a perfect model test, to be well calibrated for combining, as well as evaluating separately, independent evidence (Lewis 2018). That is, the uncertainty ranges output by these two methods, although different in statistical nature, are both close to exact confidence intervals. Accordingly, in the long run probabilistic conclusions by an investigator employing either of these methods will on average be true statements, which is surely highly desirable for scientific investigations. That is not in general the case for subjective Bayesian methods (Fraser 2011, Lewis 2014).
Moreover, Bayesian updating does not in general produce satisfactorily calibrated inference when combining evidence, even if the related Bayesian inference from the separate pieces of evidence is well calibrated (Lewis 2013, Lewis 2018). Nor is Bayesian updating satisfactory as a method of incorporating probabilistic prior information, which can however be incorporated under the aforementioned objective Bayesian method. The appropriate way to do so is by treating the prior information not as a prior density to be used in Bayesian updating, but as equivalent to a notional observation with a certain probability density, from which a posterior density has been calculated using Bayes' theorem with a noninformative prior (Hartigan 1965).
In order to achieve satisfactory inference about climate sensitivity when combining evidence, climate scientists need to move on from fundamentally flawed subjective Bayesian methods, and to cease ignoring the existence of objective Bayesian and frequentist (profile) likelihood ratio based methods that are both demonstrably superior.

A: Besides the title of "objective" Bayesian method, which we find confusing and misleading, there are several reasons why such methods are not investigated nor acknowledged in this study. One of the first reasons would be that this paper does not aim at being a summary of every possible Bayesian method, but solely introduces one method for the question of emergent constraints in comparison to other (non-Bayesian) methods used in the past.
Having said so, "non-informative" prior such as Jeffrey's prior, are, in fact, very informative when dealing with a single problem which carry information by itself, such as the relationship between Sensitivity (S) and Temperature (T) in a defined ensemble of climate model. Actually, we do consider that informative priors are a valuable advantage of Bayesian methods (or updating), as it carries the knowledge, with a given uncertainty, of the original problem (in that case, the plausible range of S). There is no reason for thinking that one prior would be more non-informative than others in every case - priors are more informative than others based on the problem. In the

case of this paper, we could consider that the uniform prior is more informative than the Cauchy prior towards high S. However, such affirmation could be completely different with a different set of climate system parameters. Thus, there is no reason for thinking that one specific Bayesian method would have more or less flaws than another.

———————————————————

Short Comment #2

A: The primary criticism of S&W is that our underlying model is different to that adopted in previous Emergent Constraint (EC) research. We agree that we are presenting a fundamental and perhaps radical point of departure from previous work and we will emphasise this more clearly in the text. In our view, it is entirely natural when attempting to estimate a parameter such as ECS, that we take a prior on ECS which is explicitly stated, and then ask how the evidence under consideration (in this example, a temperature observation and an ensemble of GCMs that exhibit a relationship between their ECS values and this temperature) can be used to update this prior. We believe that our approach, while unlike much previous work in EC, is actually much more closely aligned with the bulk of the research on Bayesian estimation of ECS, and our method allows for these data to be easily used to update any prior on ECS that might arise from a previous unrelated study. In contrast, as Sansom and Williamson showed so elegantly themselves, a Bayesian formulation of standard EC approach would be to take a prior over the measurand and infer ECS as a diagnostic of this. While this seems to work well for their "reference prior" case and they also discuss physically-motivated priors on the regression parameters, it does not seem so clear (and they do not discuss) alternative priors for the measurand and the implied predictive prior for ECS. Of course we do not insist that our view is the only correct one and must be adopted by others, but we believe strongly that it is reasonable and worthy of serious consideration.

C: The central feature of the proposed approach is to specify $p(X_\star \mid Y_\star)$ and $p(X \mid Y)$ rather than $p(Y_\star \mid X_\star)$ and $p(Y \mid X)$. It is important to realise that these are two fundamentally different statistical models, they cannot both be valid at the same time, and will inevitably result in different inferences for $Y$. Though both equations hold mathematically (as they are valid factorisations of the joint distribution), they imply different conditional independencies in the modelling that need to be physically interpreted and are certainly not interchangable. Therefore, one must consider carefully the underlying reasoning before adopting one or the other. For emergent constraints, $Y_\star$ is usually a measurable property of the future climate and $X_\star$ an observable property of the current or historical climate. Therefore it makes immediate sense to adopt the standard model for emergent constraints, i.e., the future depends upon the past via $p(Y_\star \mid X_\star)$ and $p(Y \mid X)$. In those cases the proposed approach would make no sense because it explicitly states that the past depends on the future via $p(X_\star \mid Y_\star)$ and $p(X \mid Y)$. Equilibrium Climate Sensitivity (ECS) is operationally defined as "the temperature anomaly reached at equilibrium following a instantaneous doubling of CO2". To us, it seems natural to view this as a future climate quantity, and so it makes sense to adopt the standard model for ECS.
It might be possible to justify the proposed model for certain quantities of interest, but for an operationally defined future quantity, we would have to be willing to accept the reversal of time's arrow, i.e., past depends on future. It may be that the authors have in mind some alternative definition or interpretation of ECS that renders time's arrow an illusion for this quantity in particular.

A: We have to disagree with this comment for several reasons. In the definition of emergent constraints, there is strictly no assumption that one variable belongs to the future, nor that another variable belongs to the past. The theory of emergent constraints is solely based on a physically plausible relationship between two variables of the climate system. Having said so, Sansom & Williamson argue that ECS should be viewed as a future climate quantity, which is not the view we share. We interpret ECS as a parameter of the climate system which is independent of time. In particular, it is usually accepted that small forcing, such as the traditional 4 times CO2 forcing used to compute ECS leads to weak non-linearity. Therefore, ECS can be approximated as a constant parameter of the climate system under a certain forcing, and has its own existence in the present time.

As written earlier for Reviewer #2, we added the following: "Although the unobserved variable is usually taken as a future variable, the emergent constraints theory can be used with two variables within the same time frame, as long as the relationship is plausible."

We think this is a frequent source of confusion, when there is no restriction on the emergent constraint theory to be used with one past and one future variable. We also added physical arguments (see following comment) which explicitly states that ECS computed from 4xCO2 experiment usually leads to weak non-linearity, and thus can be considered as a constant parameter of the climate system under certain forcing and be used within the LGM and mPWP framework.

C: we must strongly insist that the physical argument behind the statistical model be made explicit, well defended, and open to the scrutiny of other researchers within the field.

A: It is rather straightforward to find arguments for the two models. One way (where S is the predicted and T the predictand) was already introduced by Annan and Hargreaves (2016). Here, we stipulate that, with a certain knowledge about S, one could estimate a value of tropical T, accounting for a given uncertainty coming from temperature changes non-related to S. Physical arguments have been added in the revised paper, as the following:

"It implies that S is able to give a prediction of T_tropical, with a given uncertainty. This is physically plausible, as S is considered as one of the best metric to represent temperature change. In particular, S is often diagnosed in climate models from abrupt and sustained quadrupling of $CO_2$ from pre-industrial conditions ($4xCO_2$), which usually leads to weak non-linearity similar to what shall be observed from LGM or mPWP climate dynamics. Therefore, it is possible to use $4xCO_2$-computed S of climate models to predict T_tropical, assuming epsilon as a representation of all processes not related to S."

C: However the statistical model is not stated explicitly in its entirety, nor are the models it is compared to. For the sake of transparency, we interpret the model used by the authors to be
$X_m \sim Normal(\beta_0 + \beta_1 Y_m, \sigma^2)$ form=1,...,M
$X_\star \sim Normal(\beta_0 + \beta_1 Y_\star, \sigma^2)$
$Z \sim Normal(X_\star, \sigma_Z^2)$
Examination of those same Python scripts reveals that four chains of only 2 000 samples each with no warm-up were used in the production of the reported results (a total of only 8 000 samples). Both STAN and PyMC3 (used by the authors) implement the No U-Turn Sampler (NUTS) variant of Hamiltonian Monte Carlo for efficient mixing and fast convergence, so our results should be comparable. The lack of warm-up / burn-in period used by the authors is likely to lead to skewed estimates, even using NUTS. Further, the inefficient use of importance sampling to account for the observation uncertainty and the small total number of samples given notorious the difficulty of efficiently sampling from the Cauchy distribution are all likely to contribute to the differences we see when implementing their framework. Unless we have misunderstood their modelling (and by itself this would be an argument for making it explicit in the manuscript), we are not sure that the numbers given in the text actually represent the posteriors of their alternative model faithfully.

A: There is a difference between the model above (which is a simple linear regression) and the model used in the paper. This is mainly due to a lack of clarification of the model as already pointed out by Reviewer #2 which has been corrected. Therefore, the non-reproducibility of the results shown by Williamson and Sansom is not due the amount of samples as commented here. This is mainly due to a mistake in the code which was posted online. All computations were performed with 200 000 samples (and not 2000) for a total of 800 000 samples. By default, the code came with 2000 samples for faster simulation, but was not meant to be published with this number. This mistake has been corrected and we thank the authors of the comment to have spotted it. Additionally, this comment made us think about the size of burn-in period, which has been increased. To check the convergence of the chains, each code comes with a Gelman-Rubin test, easily implemented by PyMC3.

Having said so, the model has been made more explicit as written above in the response to Reviewer #2. Additionally, the code provided has been changed to make them efficient but also slightly faster to run. By default, the BLR produces 100,000 samples (400,000 with 4 chains), and

burn-in 10,000 samples. Importance sampling uses 800,000 samples. NUTS (PyMC3) comes by default with a substantial burn-in period, which is not originally written.

C: Bowman et al. (2018) include explicit expressions for projection using the Kalman filter model (Equations 17 & 23). However using these expressions we are also unable to reproduce the results for the Kalman filter quoted in Table 2 of the manuscript. Examining the Python script that accompanies this manuscript reveals an obvious error in the expression for the posterior variance on Line 80. The authors should therefore revisit the calculation of these credible intervals.

A: The code has been corrected, as a matrix multiplication was missing. The direct consequence of this is having a more constrained posterior range using the Kalman Filter (roughly 0.5 K more constrained on each bounds), which tend to show that the Kalman Filter is even more restrictive than expected.

C: We were able to reproduce the Ordinary Least Squares (OLS) estimates and intervals. However, in Section 2.1 the authors equate OLS with frequentist linear regression. This is incorrect. As discussed in detail in Williamson & Sansom (2019), OLS is a purely algorithmic method of parameter estimation in a mathematical model. The OLS estimates of the mean parameters in a linear regression model are equal to those obtained by frequentist maximum likelihood estimation, but OLS provides no estimate of uncertainty in either the parameters or the prediction. Frequentist regression is difficult to justify in a climate change context, but Bracegirdle & Stephenson (2011) presented emergent constraints within a frequentist linear regression framework and this approach has been adopted in many subsequent studies. However, the authors present heuristic uncertainty estimates based on mean-squared errors and on lines 287-288 claim that "OLS" underestimates uncertainty compared to their method. In fact, when standard frequentist regression is used, the inference for ECS in the LGM PMIP2 experiment is very similar to the proposed model with median 2.8K and 90% confidence interval 1.0–4.5K. As Williamson & Sansom (2019) point out, this is equal to an equal tailed 90% Bayesian credible interval under reference priors. Credible intervals from the reference model under the other experiments are less similar to the proposed model, but wider than either the "OLS" or Kalman filter estimates.

A: We might have taken a non-elegant shortcut here. The main idea was that OLS, at the opposite of Bayesian regression, does not require any prior knowledge, which makes it closer to frequentist philosophy. Nevertheless, this does not exclude OLS of being one of the most used method and therefore has a large interest for multi-study comparison. Our terminology may also have been a little sloppy. While we agree OLS per se does not provide uncertainty estimates, these are readily obtained from standard regression methods which are used for the OLS estimate. Finally, the uncertainty estimates were updated following the method of generating a predictive ensemble, as they are computed in the Bayesian methods and as they were computed in previous studies (Hargreaves et al., 2012; Schmidt et al., 2014; Hargreaves and Annan, 2016). As shown by the authors of the comment, the new intervals are slightly wider than the previous ones, but still much tighter than the Bayesian intervals in the case of low correlation, which was already the original argument of this method comparison.
We changed the terminology of OLS since it does not produce confidence intervals, but statistical models based on it will produce these intervals.

C: Treatment of model inadequacy
In the statistical reasoning above we omitted discussion of model inadequacy for brevity. In lines 130–134 and lines 376–378 the authors claim that in their approach model inadequacy can be entirely accounted for by specifying a larger residual variance for reality than the models and it is not necessary to consider differences in the regression parameters. With a minor modification to the proposed model, the intercept can be made independent of the slope, and therefore any additional uncertainty about the intercept in the real world can safely be pushed into the residual since both sources⋆ of uncertainty are independent of Y . However, any uncertainty in the slope leads to uncertainty about X⋆ that is dependent on Y ⋆, i.e, the width of the predictive interval for X⋆ increases with the distance of Y ⋆ from the prior mean. Therefore any additional uncertainty in the slope in the real world is also conditional on the value of Y ⋆ and not accounted for by the residual variance. This is a simple matter of geometry and is therefore unavoidable. Williamson and Sansom (2019) developed a coherent elicitation of the regression parameters and structural error that was designed to account for these geometric considerations, and any emergent

constraints framework that wishes to account for structural error, whether using the authors approach or the standard one, must grapple with the geometry.

A: We disagree with this comment. Although accounting for model inadequacy is relevant to the question of emergent constraints, accounting for it on every regression parameters leads to creation of regression lines based on an ensemble of non-existing models. We are aware of the Williamson and Sansom approach which postulates a different regression line based on some hypothetical ensemble of future models, and while we agree with them that treatment of model inadequacy is important we don't find their approach entirely compelling. In reality, there is one sensitivity value and one temperature observation, and it does not seem helpful to us to use 3 additional parameters including a new regression line to describe this. In the limit of improved models (W&S Section 4b), we would expect them all to converge to the truth and it is not clear that a regression line and error term is particularly meaningful in this case. Furthermore, it might be reasonable to expect that in this limiting ensemble $sigma^*$ should be rather smaller, instead of larger, than sigma for the existing ensemble.

We emphasise that our point here is not to criticise W&S who have made a set of judgments that they consider reasonable, but merely to explain why ours differ.

We think this discussion is somewhat beyond the scope of our study, which is focused on the use of an existing ensemble of climate models.

[revised manuscript text omitted]